# FERRET: REFER AND GROUND ANYTHING ANYWHERE AT ANY GRANULARITY

🍎Haoxuan You[1][†], Haotian Zhang[2][†], Zhe Gan[2], Xianzhi Du[2], Bowen Zhang[2], Zirui Wang[2], Liangliang Cao[2], Shih-Fu Chang[1], Yinfei Yang[2]
[1]Columbia University, [2]Apple AI/ML
haoxuan.you@cs.columbia.edu, {haotian_zhang2,zhe.gan,yinfeiy}@apple.com

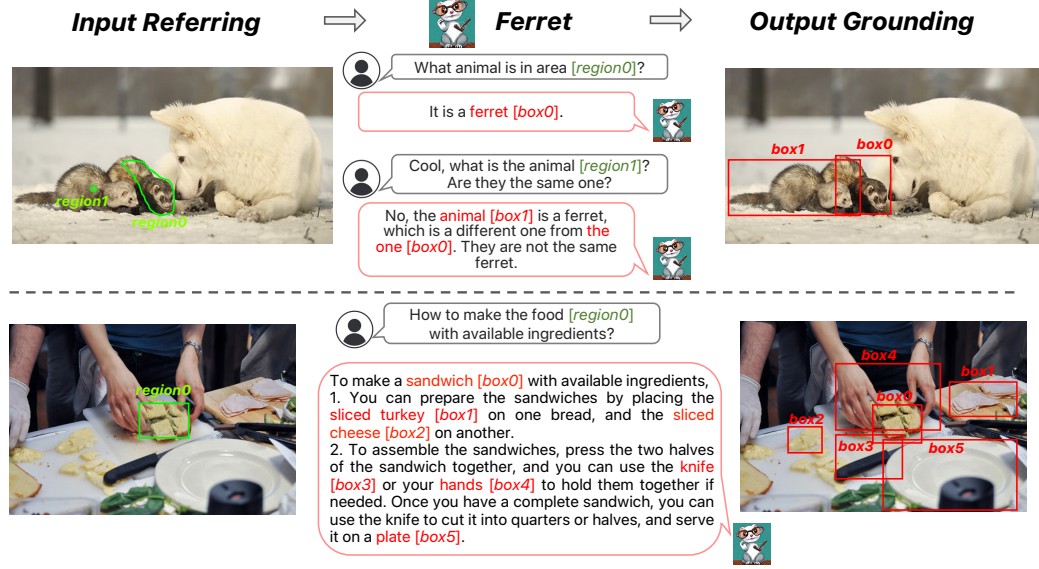

Figure 1: Ferret enables *referring* and *grounding* capabilities for multimodal large language model (LLM). In terms of referring, a user can refer to a region or an object in point, box, or any free-form shape. The *regionN* in the input will be replaced by the proposed hybrid representation before being fed into the LLM. In terms of grounding, Ferret is able to accurately ground any open-vocabulary descriptions. The *boxN* in the output denotes the predicted bounding box coordinates.

## ABSTRACT

We introduce Ferret, a new Multimodal Large Language Model (MLLM) capable of understanding spatial referring of any shape or granularity within an image and accurately grounding open-vocabulary descriptions. To unify referring and grounding in the LLM paradigm, Ferret employs a novel and powerful hybrid region representation that integrates discrete coordinates and continuous features jointly to represent a region in the image. To extract the continuous features of versatile regions, we propose a spatial-aware visual sampler, adept at handling varying sparsity across different shapes. Consequently, Ferret can accept diverse region inputs, such as points, bounding boxes, and free-form shapes. To bolster the desired capability of Ferret, we curate GRIT, a comprehensive refer-and-ground instruction tuning dataset including 1.1M samples that contain rich hierarchical spatial knowledge, with 95K hard negative data to promote model robustness. The resulting model not only achieves superior performance in classical referring and grounding tasks, but also greatly outperforms existing MLLMs in region-based and localization-demanded multimodal chatting. Our evaluations also reveal a significantly improved capability of describing image details and a remarkable alleviation in object hallucination. Code and data are available at https://github.com/apple/ml-ferret.

---

🍎Work done during an internship at Apple. †Equal contribution.

# 1 INTRODUCTION

In vision-language learning, how to enable spatial understanding in models is a fundamental research problem. Two desired capabilities stem from this problem: *referring* and *grounding*. Referring demands that the model can accurately comprehend the semantics of specific given regions (Krahmer & Van Deemter, 2012; Kazemzadeh et al., 2014; Mao et al., 2016; Yu et al., 2016; Zellers et al., 2019), whereas grounding necessitates that the model to localize the region in accordance with the given semantic description (Luo & Shakhnarovich, 2017; Nagaraja et al., 2016; Yu et al., 2017; Kamath et al., 2021).

Essentially, *referring* and *grounding* demand the same type of knowledge: alignment of spatial information and semantics. Despite this, existing works mostly learn referring and grounding individually (Li et al., 2022; Wu et al., 2022; Yu et al., 2017). In comparison, humans can learn from one task and generalize the shared knowledge to the other task effortlessly, and are able to seamlessly integrate referring/grounding capabilities with daily dialogue and reasoning (Zellers et al., 2019). Inspired by the above gap, in this paper, we study three main questions: (*i*) How to unify referring and grounding in one framework, and will they benefit each other? (*ii*) How to represent versatile types of regions that humans usually use for referring, such as point, box, scribble, and even free-form shapes? (*iii*) How to make referring and grounding open-vocabulary, instruction-following, and robust, which are crucial for practical applications?

Targeting these three questions, we introduce **Ferret**, a novel refer-and-ground Multimodal Large Language Model (MLLM). First of all, we choose MLLM as the bedrock of Ferret to leverage their powerful vision-language global understanding capability (Zhu et al., 2023a; Liu et al., 2023b; Li et al., 2023c). To unify referring and grounding, Ferret first represents the coordinates of regions in natural language numerical form,[1] as illustrated in Figure 3. However, it is inefficient to use single point or box coordinates to represent versatile shapes of regions, such as strokes, scribbles, or complex polygons. These shapes are essential for more universal and precise human-model interaction. To solve this problem, we further propose a spatial-aware visual sampler to acquire the visual features for regions in any shape, taking care of the varying sparsity in those shapes. Then, the discrete coordinates and the continuous visual features are combined together to represent the visual regions in the input, composing a hybrid region representation in Ferret. Equipped with above methods, Ferret can deal with input that mixes referred regions with free-form text, and is able to ground the mentioned objects in its output by seamlessly generating the coordinates for each groundable object along with generating text. To our best knowledge, Ferret is the first work that is able to process free-formed region inputs in MLLMs.

In order to make the refer-and-ground capability in Ferret open-vocabulary, instruction-following, and robust, we collect **GRIT**, a **G**round-and-**R**efer **I**nstruction-**T**uning dataset with 1.1M samples. GRIT contains multiple levels of spatial knowledge, covering objects, relationships, region descriptions, and complex reasoning. It includes both text-in location-out (grounding) and location-in text-out (referring) data, as well as data that mixes location and text in both input and output. The majority of the dataset is converted from existing vision(-language) tasks like object detection (Krishna et al., 2017) and phrase grounding (Yu et al., 2016; Plummer et al., 2015) with carefully designed templates to make it instruction-following. Additionally, 34K refer-and-ground instruction-tuning conversations are collected via the help of ChatGPT/GPT-4 (OpenAI, 2023b) to facilitate training an instruction-following and open-vocabulary refer-and-ground generalist. Moreover, we conduct spatial-aware negative data mining, which further promotes model robustness.

Ferret subsumes strong open-vocabulary capabilities of spatial understanding and localization. When evaluated on conventional referring and grounding tasks, it achieves superior performance. More than that, we believe refer-and-ground capabilities should be integrated into daily conversations of humans, *e.g.*, people refer to something they don't know and ask what it is used for (like Figure 1). To evaluate this new capability, we introduce **Ferret-Bench**, covering three new types of tasks: Referring Description, Referring Reasoning, and Grounding in Conversation. We benchmark existing MLLMs and observe that Ferret can outperform the best of them by 20.4% on average. Moreover, Ferret demonstrates an intriguing property of alleviating object hallucinations.

In summary, our contributions are threefold. (*i*) We propose Ferret, that uses a hybrid region representation equipped with a novel spatial-aware visual sampler, to enable fine-grained and open-

---

[1]Note that there is no additional vocabulary or position encoders introduced in Ferret model.

vocabulary referring and grounding in MLLM. (*ii*) We construct GRIT, a large-scale ground-and-refer instruction tuning dataset, for model training. It also contains additional spatial negative samples to enhance model robustness. (*iii*) We introduce Ferret-Bench, to evaluate tasks jointly requiring referring/grounding, semantics, knowledge, and reasoning. Our model exhibits superior performance in a wide range of tasks and reduces object hallucination.

## 2 METHOD

We start with detailing the proposed hybrid region representation to depict regions of various shapes and formats. Then, we present the model architecture of Ferret.

### 2.1 HYBRID REGION REPRESENTATION

When referring to specific regions, three primary formats are generally used: point, box, and free-form shapes. While the point and box formats can be succinctly represented by coordinates (*e.g.*, $[x, y]$ for a point, and $[x_{\min}, y_{\min}, x_{\max}, y_{\max}]$ for a box) as in Peng et al. (2023); Chen et al. (2023b), the free-form shape is more versatile, encompassing a variety of region types such as scribbles, polygons, and masks. The advantage of free-form shape is straightforwardly illustrated in Figure 2. Depicting free-form shapes through coordinates is computationally expensive and obscure, and its complexity hinders the model learning to establish a clear correlation between the provided coordinates and the corresponding regions.

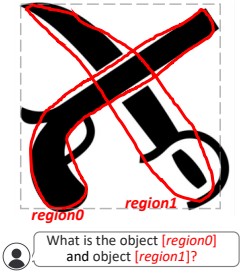

Figure 2: Bounding box *v.s.* Free-from Shape. These two objects have almost the same bounding box, causing ambiguity when relying on the box to refer to. Equipped with hybrid region representation, Ferret can separate them.

To generalize across all three distinct formats, we propose a hybrid region representation that synergizes discrete coordinates with continuous visual features to refer to a particular region, which is shown in the top-left of Figure 3. For coordinates, following Chen et al. (2021); Yang et al. (2022), we quantize each coordinate into one of the $n_{\text{bins}}$ discrete bins.[2] Regarding continuous visual features, for a given region $\mathbf{R}$, we first construct a 2D binary mask $\mathbf{M}$ of the same size as the image, marking a value of 1 inside the targeted region and 0 outside of the region. Then, the binary mask $\mathbf{M}$, jointly with the extracted image feature map $\mathbf{Z}$, is sent into our proposed spatial-aware visual sampler $s(\cdot)$, which will be detailed in Section 2.2, to extract the visual continuous feature $\mathbf{f} = s(\mathbf{M}, \mathbf{Z})$.

Finally, we represent a point with $\{x, y, \mathbf{f}_{R_p}\}$, where the region $R_p$ is a circle centered in $\{x, y\}$ with a fixed radius.[3] A box or a free-form shape can both be represented by $\{x_{\min}, y_{\min}, x_{\max}, y_{\max}, \mathbf{f}_{R_{box}}\}$, where $x_{\min}/x_{\max}$ denotes the minimum/maximum $x$-axis coordinate of the region, and so forth for $y$-axis. $R_{box}$ denotes the input region.

### 2.2 MODEL ARCHITECTURE

As illustrated in Figure 3, Ferret is mainly composed of (*i*) an image encoder to extract image embeddings, (*ii*) the proposed spatial-aware visual sampler to extract regional continuous features, and (*iii*) an LLM to jointly model image, text, and region features.

**Input.** We feed the image into a pre-trained visual encoder, CLIP-ViT-L/14 (Radford et al., 2021), to extract the image embeddings $\mathbf{Z} \in \mathbb{R}^{H \times W \times C}$. For text, we tokenize the text sequence using the pre-trained LLM's tokenizer and project them into text embeddings $\mathbf{T} \in \mathbb{R}^{L \times D}$. As for referred regions, we append the coordinates and a special token as a placeholder for continuous features after the name of the region: "⟨region_name⟩ ⟨coordinates⟩ ⟨SPE⟩". For example, "a cat [100, 50, 200, 300] ⟨SPE⟩". If the name is unknown or hard to describe because multiple objects are included, we just use "region" or "area" as the "⟨region_name⟩". In this way, referred regions can be well mixed with ordinary texts to form complete sentences.

---

[2]$n_{\text{bins}} = 1000$ by default. The value is input invariant, which means for any input image size, the original coordinate will be mapped to the new coordinates. This makes the model robust to different input resolutions.

[3]Radius is set to 5 by default.

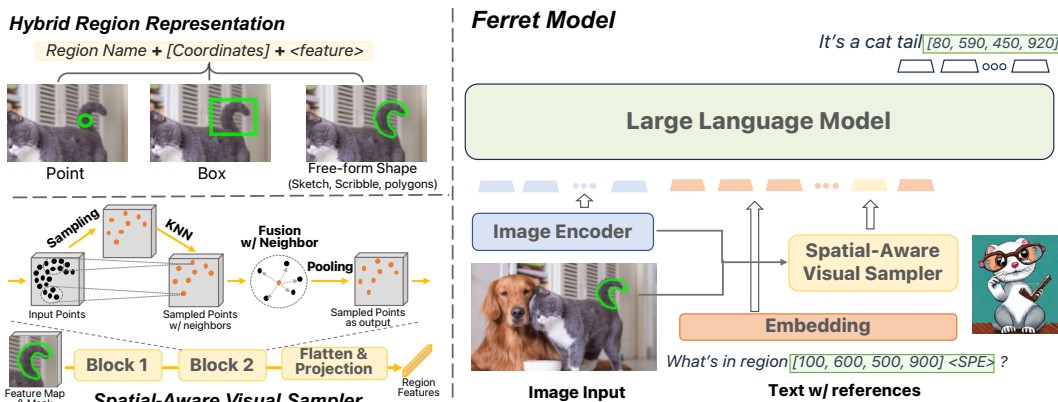

Figure 3: Overview of the proposed Ferret model architecture. (Left) The proposed hybrid region representation and spatial-aware visual sampler. (Right) Overall model architecture. All parameters besides the image encoder are trainable.

**Spatial-aware Visual Sampler.** The shape of the referred regions can be quite varied, not limited to just points or rectangle boxes. Grid-based processing like convolution or patch attention cannot handle irregular shapes. Similar to our cases, 3D point clouds are also in irregular shape and show varied sparsity in the 3D space. Inspired by existing works in 3D point cloud learning (Qi et al., 2017a; Ma et al., 2022; Wang et al., 2019), we propose a spatial-aware visual sampler.

Given extracted image feature map $\mathbf{Z} \in \mathbb{R}^{H \times W \times C}$ and the binary region mask $\mathbf{M}$, we first randomly sample $N$ positive points inside $\mathbf{M}$. For each point, its feature is obtained by bilinear interpolation. The $N$ points are fed into a cascade of blocks, where each of them includes three steps: sampling, gathering, pooling. (1) Sampling: $\frac{N}{r}$ points are sampled from $N$ points via farthest point sampling (FPS) algorithm (Qi et al., 2017b),[4] which can guarantee sufficient coverage. (2) Gathering: For each of the sampled points $x_i$, we search its $k$ nearest neighbors from the pool of previous $N$ points, and obtain a group of points $\{x_{i1}, x_{i2}, ..., x_{ik}\}$. Then, inspired by PointMLP (Ma et al., 2022), for each group, we fuse the features of sampled point $x_i$ and it neighbor points by:

$$h_{ik} = \sigma([\theta([\mathbf{Z}(x_{ik}) - \mathbf{Z}(x_i); C(x_{ik}) - C(x_i)]); \mathbf{Z}(x_i); C(x_i)]) , \qquad (1)$$

where $x_{ik}$ is one of the neighbors of $x_i$, $\mathbf{Z}(x)$ denotes the point $x$'s feature (in the first block, it is interpolated from feature map $\mathbf{Z}$; in the succeeding blocks, it is the output feature from the previous block), $C(x)$ denotes the 2D coordinates of point $x$, $[;]$ means channel-wise concatenation of multiple vectors, $\theta$ is implemented by a linear layer to adapt the relative local features, and $\sigma$ is also a linear layer to fuse each local feature from neighbors with sampled point feature. (3) Pooling: A max pooling is conducted to fuse $k$ neighbor features into one feature as the representation of the sampled point:

$$h_i = \max_{k:(x_{ik}) \in \text{KNNs of } x_i} h_{ik} . \qquad (2)$$

After the three steps, we obtain fewer points but a more dense feature space since it incorporates the local neighbor features as well as their relative positions. In experiments, we set $N$=512, $r$=4 and $k$=24, and cascade two such blocks, which in the end outputs 32 points with their features. Similar to ROIAlign (He et al., 2017), we flatten the point features into a single vector and project it to the dimension of LLM embeddings. The final feature is used to replace the $\langle \text{SPE} \rangle$ token in the input.

**Output.** The above region denotations are used in Ferret input to refer to specific regions. In Ferret output, to achieve grounding, we generate the box coordinates right after the corresponding regions/nouns in the text response. For instance, "There is a dog [100, 150, 300, 200] in the figure." With this data format, our model is expected to implicitly learn what is groundable in the current image and what their locations are.

**LLM.** We consider Vicuna (Chiang et al., 2023) as our language model, a decoder-only LLM (Brown et al., 2020) that is instruction-tuned on top of LLaMA (Touvron et al., 2023a). Prior to being fed into the LLM, the image embeddings undergo transformation via an additional linear layer to match the embedding dimension of the text tokens.

[4]FPS starts from a random single point sampled from $N$ points. In each iteration, it samples one point from the rest points such that it is the farthest from the set of already sampled points. See detail in Qi et al. (2017b).

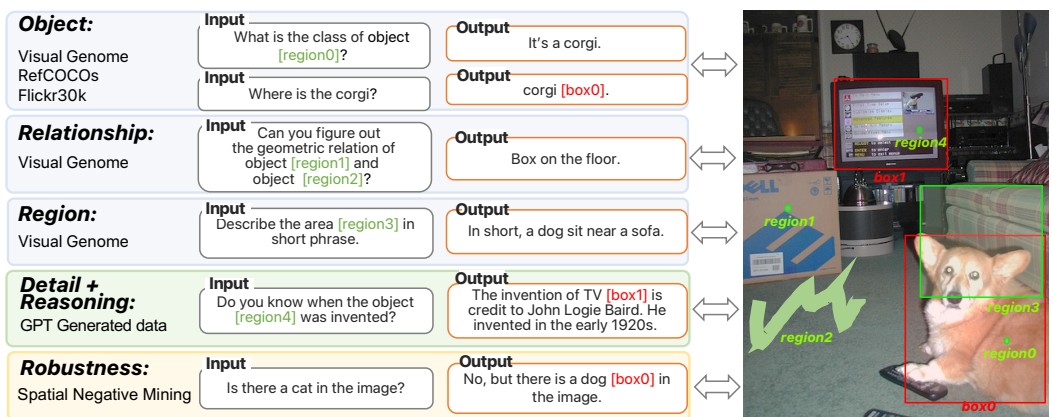

Figure 4: Overview of the GRIT dataset for Ferret model training. It contains three types of data: ($i$) public datasets that are converted into an instruction-following format (the top-3 rows); ($ii$) data generated via prompting ChatGPT and GPT-4 (the 4th row); and ($iii$) negative data to enhance model robustness (the last row).

## 3 GRIT: GROUND-AND-REFER INSTRUCTION-TUNING DATASET

In this section, we present GRIT, a **G**round-and-**R**efer **I**nstruction-**T**uning dataset containing around 1.1M multimodal dialogues for model training. GRIT consists of three types of data: ($i$) public datasets that are converted into an instruction-following format (Section 3.1); ($ii$) instruction-tuning data generated via ChatGPT and GPT-4 (Section 3.2); and ($iii$) additional data from spatial negative mining for enhancing model robustness (Section 3.3).

### 3.1 HIERARCHY

Spatial understanding can be characterized by varying levels of granularity and task formats. During our dataset creation, we look into the following categories based on two dimensions:

- In terms of *granularity*, we identify four main categories: ($i$) individual objects, ($ii$) relationships among objects, ($iii$) descriptions of specific regions, and ($iv$) region-based complex reasoning.
- In terms of *task format*, we further divide the data into three distinct types: ($i$) Region-in Text-out data, ($ii$) Text-in Region-out data, and ($iii$) Text-Region combined data.[5]

We compiled an extensive set of public data focusing on the aforementioned dimensions and converted them into an instruction-following format using carefully designed templates. A more in-depth view of these templates is available in Appendix C.1.

**Individual objects.** To achieve visual understanding at the object level, we select object detection datasets such as Visual Genome (Krishna et al., 2017), Object365 (Shao et al., 2019), and visual grounding datasets including RefCOCOs (Yu et al., 2016; Lin et al., 2014; Nagaraja et al., 2016) and Flickr30k-Entities (Plummer et al., 2015). The converted Visual Genome object data follow a *Region-in Text-out* format. Additionally, to enable Ferret to understand free-form shapes, we apply SAM (Kirillov et al., 2023) to Visual Genome object data to obtain a segmentation mask for each object, which is fed into the spatial-aware visual sampler to extract continuous region feature during training. The visual grounding datasets and Object365 data adhere to a *Text-in Region-out* format. This section has in total 678k data.

**Relationships among objects & descriptions of regions.** We selected data pertaining to object relationships and region captions from Visual Genome (Krishna et al., 2017) to address these two facets, respectively. Both datasets employ a *Region-in Text-out* format and 177k data are obtained. Similar to Visual Genome object data, we also extract segmentation masks of objects in Visual Genome relationship data via SAM.

---

[5]For Region-in Text-out data, the input highlights a specific region, prompting queries about it. For Text-in Region-out data, the input comprises textual descriptions, and the task is to pinpoint or ground the relevant region in its response. The combined Text-Region data integrates both text and region within a single sequence, which can be present in the input, output, or both.

**Region-based complex reasoning.** Regarding complex reasoning centered on specific regions, we constructed a novel dataset with the help of ChatGPT/GPT-4. It adopts a combined Text-Region format, and is detailed in the subsequent section.

## 3.2 GPT-ASSISTED VISUAL INSTRUCTION DATA GENERATION

Besides converting existing datasets by templates, dialogue instruction tuning data is proved to be critical for MLLM to understand human intention and generate fluent, natural, and long-form responses (Liu et al., 2023b; Zhu et al., 2023a; Li et al., 2023d). Few-shot prompting is widely used to obtain visual instruction tuning data, where textual scene descriptions of images and human-annotated dialogues are provided as few-shot demonstrations, and ChatGPT/GPT4 are prompted to generate new dialogue based on the new image's textual scene descriptions.

However, previous instruction tuning data mainly focus on describing the entire image without explicitly specifying spatial-related information. To collect refer-and-ground instruction tuning data, we emphasize region-based spatial knowledge in the following three steps. ($i$) Besides objects and global captions usually used as before, our symbolic scene description additionally includes physical relationships between objects and region captions along with coordinates of them. ($ii$) In human-annotated dialogues, we add coordinates after the groundable regions or objects either in input or output or both, and the dialogues are typically focused on specific regions. It helps to implicitly prompt ChatGPT/GPT4 to follow similar patterns when generating new dialogues. ($iii$) The generated dialogues sometimes cannot follow the rules and patterns we wrote in system prompts and few-shot examples, which might be due to that the context of LLM input is too long to handle all the details. To alleviate it, we propose to use ChatGPT/GPT-4 again to refine the initially generated dialogues, whose context length is only 10% of the data generated from the first round on average. To save cost, we use ChatGPT in the first round of generation and GPT-4 for refining. 34k dialogues in total are collected.

Additionally, to exploit existing instruction-tuning data such as those in LLaVA (Liu et al., 2023b), we apply an open-vocabulary object detector, GLIPv2 (Zhang et al., 2022), on LLaVA-158k data to localize groundable nouns in the text. Then, we append the bounding boxes after the corresponding nouns, forming a pseudo-grounded LLaVA instruction data that are also used for training Ferret.

## 3.3 SPATIAL NEGATIVE MINING

As highlighted in prior studies (Li et al., 2023e; Liu et al., 2023a), MLLM exhibits a propensity to hallucinate in response to yes/no questions. We observed a similar occurrence when inquiring about detailed regions. To address this, we also conduct negative sample mining by following two ways: ($i$) *Image-conditioned Category Localization*, and ($ii$) *Semantics-conditioned Category Localization*. They both ask the model to localize specific object categories, thereby enabling the model's ability to discern and potentially recognize the absence of certain objects. They differ in how to select the negative category. For ($i$), Object365 data are employed and we randomly select the object class from the vocabulary that is not shown in the given image. For ($ii$), Flickr30k data are used and negative categories are sourced by utilizing ChatGPT/GPT4 to find entities that are most analogous to the original class, attribute, or quantity, *e.g.*, 'man' vs. 'woman', 'blue' vs. 'yellow', 'two' vs. 'three'.

We curate the data to maintain an equilibrium between positive and negative samples for each of the two types.[6] 95k data are collected. A more comprehensive elaboration is provided in Appendix C.2.

## 4 EXPERIMENTS

First of all, we illustrate the training details of Ferret. Then in evaluation, we start with evaluating Ferret on conventional referring and grounding benchmarks (Sec. 4.1 and 4.2). Then, we demonstrate the power of Ferret in more complex multimodal chatting with refer-and-ground capability in Sec. 4.3. For a detailed visualization of each, kindly check Appendix E. We further ablate key components in Ferret (Sec. 4.4), analyze the object hallucination of Ferret (Sec. 4.5) and discuss Ferret *v.s.* GPT-4V (Sec. **??**).

---

[6]We observed that even though we don't collect other data specifically for training, Ferret demonstrates the capability to generalize robustness across diverse categories like relationships, events, *etc*. We attribute this versatility to the potent compositional capabilities inherent to LLM.

Table 1: Results of referring object classification on three different referring types, including point, box, and free-form shape. '✗' means no such capability.

| Models | LVIS (Acc %) | | |
|---|---|---|---|
| | Point | Box | Free-form |
| Random Guess | 50 | 50 | 50 |
| LLaVA | 50.1 | 50.3 | ✗ |
| Kosmos-2 (Peng et al., 2023) | ✗ | 60.25 | ✗ |
| Shikra-7B (Chen et al., 2023b) | 57.82 | 67.71 | ✗ |
| GPT4-ROI (Zhang et al., 2023) | ✗ | 61.76 | ✗ |
| Ferret-7B | 67.94 | 79.42 | 69.77 |
| Ferret-13B | **68.35** | **80.46** | **70.98** |

Table 2: Results of grounded image captioning on the test set of Flickr30k Entities. BLEU@4, METEOR, CIDEr, and SPICE are used for the caption evaluation. $F1_{all}$ and $F1_{loc}$ are used for grounding evaluation. '–' means not reported.

| Models | Caption Eval. | | | | Grounding Eval. | |
|---|---|---|---|---|---|---|
| | B@4 | M | C | S | $F1_{all}$ | $F1_{loc}$ |
| GVD (Zhou et al., 2019) | 27.3 | 22.5 | 62.3 | 16.5 | 7.55 | 22.2 |
| Cyclical (Ma et al., 2020) | 26.8 | 22.4 | 61.1 | 16.8 | 8.44 | 22.78 |
| POS-SCAN (Zhou et al., 2020) | 30.1 | 22.6 | 69.3 | 16.8 | 7.17 | 17.49 |
| UniTAB (Yang et al., 2022) | 30.1 | 23.7 | 69.7 | 17.4 | 12.95 | 34.79 |
| Shikra-13B (Chen et al., 2023b) | – | – | 73.9 | – | – | – |
| Ferret-7B | 35.1 | 24.6 | 74.8 | 18.0 | 15.02 | 37.62 |
| Ferret-13B | **37.0** | **25.5** | **76.1** | **18.3** | **15.12** | **38.03** |

**Training Details.** We initialize the image encoder with CLIP-ViT-L/14@336p, the LLM with Vicuna, and the projection layer with LLaVA's first-stage weights, leaving the visual sampler randomly initialized. After the initialization, Ferret is trained on the aforementioned GRIT data for three epochs, optimized by Loshchilov & Hutter (2017) with a learning rate of $2e-5$ and a batch size of 128. The training takes ~5/2.5 days on 8 A100 GPU for a Ferret-13B/7B. During training, when input refers to regions, we randomly choose either the center points or the bounding boxes (or segmentation masks if available) to represent the regions. We perform de-duplication in training data to remove the samples that are in downstream evaluations.

## 4.1 INPUT REFERRING

The model's capability of understanding referring is reflected in that, given a referred region in the question, how accurately the model can understand the semantics of the referred region. To measure it, we start with the most basic semantics, *object*, as it is fundamental and clear to define. To be more specific, the task we evaluate on is ***Referring Object Classification***: the question refers to a specific region in the image, and the model needs to classify the object in the region. Since Ferret and MLLMs usually generate free-form text responses, it is inaccurate to match the predicted class with the ground-truth class if directly asking the model to classify without constraints. Alternatively, we make it a binary-choice question in the format of "Is the object ⟨location⟩ a ⟨class_A⟩ or a ⟨class_B⟩?". We feed the binary-choice question and image into the MLLMs to obtain the response, and then detect if the response matches the ground-truth (GT) class by some rule.[7]

To prepare the data, we used the validation split of LVIS dataset (Gupta et al., 2019) covering over 1000 object categories, and sampled 2667 objects as the GT objects. Then, we randomly choose a different object category in the same image whose central point is close to the GT object as the negative object, and replace ⟨class_A⟩ and ⟨class_B⟩ with those two randomly to form 2667 questions. Additionally, to mimic the versatility of referring in human life, we replace the ⟨location⟩ with three different types: point, box, and free-form shape. For point, we randomly sample a point inside the GT object that is also near the GT object's boundary. For box, we use the GT bounding box provided by LVIS. For the free-form shape, we randomly generate some strokes inside the GT object to simulate that. Results on all three types of referring are summarized in Table 1. Ferret can significantly outperform previous models (Peng et al., 2023; Chen et al., 2023b) and handle all types of referring, a capability notably absent in previous works.

## 4.2 OUTPUT GROUNDING

Ferret performs well in referential dialogue, allowing for its integration into various VL tasks, notably those with grounding outputs. To rigorously assess the grounding capability, we first subject Ferret to benchmark visual grounding tasks in a generative paradigm. Then, to measure the alignments between words and regions, we further evaluate Ferret on grounded captioning task.

**Visual grounding.** Visual grounding aims to ground language queries into aligned image regions. We experiment on the sub-tasks of referring expression comprehension (REC) with three renowned benchmarks: RefCOCO (Lin et al., 2014), RefCOCO+ (Yu et al., 2016), and RefCOCOg (Mao et al., 2016), and phrase grounding with Flickr30k Entities dataset (Plummer et al., 2015). REC task involves a question or description about a specific area in an image, with the model expected to predict just one bounding box. Phrase grounding, conversely, seeks to associate all the noun

---

[7]Sometimes both GT class and negative class appear in the answer, *e.g.*, "The object is ⟨class_GT⟩, not ⟨class_Neg⟩". Our rule removes the substring in-between "not" and comma/period, and then detects GT class.

Table 3: Performance comparison (Acc@0.5) on the referring expression comprehension (RefCOCO, Ref-COCO+, RefCOCOg) and phrase grounding (Flickr30k Entities) tasks. ∗ indicates that the method is specifically fine-tuned in the second stage.

| Models | RefCOCO | | | RefCOCO+ | | | RefCOCOg | | Flickr30k Entities | |
|---|---|---|---|---|---|---|---|---|---|---|
| | val | testA | testB | val | testA | testB | val | test | val | test |
| MAttNet (Yu et al., 2018) | 76.40 | 80.43 | 69.28 | 64.93 | 70.26 | 56.00 | 66.67 | 67.01 | – | – |
| OFA-L (Wang et al., 2022b) | 79.96 | 83.67 | 76.39 | 68.29 | 76.00 | 61.75 | 67.57 | 67.58 | – | – |
| TransVG (Deng et al., 2021) | 81.02 | 82.72 | 78.35 | 64.82 | 70.70 | 56.94 | 68.67 | 67.73 | – | 79.10 |
| UNITER (Chen et al., 2020) | 81.41 | 87.04 | 74.17 | 75.90 | 81.45 | 66.70 | 74.02 | 68.67 | – | – |
| VILLA (Gan et al., 2020) | 82.39 | 87.48 | 74.84 | 76.17 | 81.54 | 66.84 | 76.18 | 76.71 | – | – |
| UniTAB (Yang et al., 2022) | 86.32 | 88.84 | 80.61 | 78.70 | 83.22 | 69.48 | 79.96 | 79.97 | 78.76 | 79.58 |
| MDETR (Kamath et al., 2021) | 86.75 | 89.58 | 81.41 | 79.52 | 84.09 | 70.62 | 81.64 | 80.89 | 82.3* | 83.8* |
| Shikra-7B (Chen et al., 2023b) | 87.01 | 90.61 | 80.24 | **81.60** | 87.36 | 72.12 | 82.27 | 82.19 | 75.84 | 76.54 |
| Ferret-7B | **87.49** | **91.35** | **82.45** | 80.78 | **87.38** | **73.14** | **83.93** | **84.76** | **80.39** | **82.21** |
| Shikra-13B (Chen et al., 2023b) | 87.83 | 91.11 | 81.81 | **82.89** | 87.79 | 74.41 | 82.64 | 83.16 | 77.41 | 78.44 |
| Ferret-13B | **89.48** | **92.41** | **84.36** | 82.81 | **88.14** | **75.17** | **85.83** | **86.34** | **81.13** | **84.76** |

phrases in the input sentence with corresponding boxes, requiring the model to predict these boxes and the word-box connections. For both tasks, we utilize uniform prompts, represented as "*What are the locations of <query>/<phrases>?*", where *<query>* denotes the textual referring expression, while *<phrases>* stands for a "comma-delimited" aggregation of the given phrases. The model is trained to output in "*<query>* [*box*]." format. The generated bounding box is considered correct if its intersection over union (IoU) with the GT box is greater than 0.5. As shown in Table 3, Ferret achieves an outstanding performance on all metrics, and is comparable to specialized fine-tuning approaches (Kamath et al., 2021).

**Grounded captioning.** The grounded captioning task requires the model to generate a caption and ground all generated noun phrases to image regions. The final predictions generally consist of three parts, *i.e.*, the text caption, visual regions as boxes, and the grounding alignments between words and boxes. Following the established benchmarks on the Flickr30k Entities dataset, we evaluate captioning and grounding separately with the captioning metrics and grounding F1 scores, respectively. $F1_{all}$ evaluates grounding as a multi-label classification problem. We also report $F1_{loc}$ that only computes the grounding score on correctly predicted object words. Results are summarized in Table 2, and Ferret achieves state-of-the-art.

### 4.3    FERRET-BENCH: MULTIMODAL CHATTING WITH REFERRING AND GROUNDING

Multimodal chatting has been an emergent ability of MLLMs. Previous benchmarks (Liu et al., 2023b) mainly evaluate conversation, detailed description, and complex reasoning via GPT-4 as a judge. Yet, a gap exists as no dataset currently evaluates multimodal chatting that necessitates referring or grounding actions, *e.g.*, instances where individuals reference an unfamiliar object and inquire about its purpose. To benchmark this intriguing and practical capability, we introduce Ferret-Bench that covers three kinds of region-based questions evaluating referring and grounding capability: ($i$) **Referring Description**: models are asked to describe a referred region based on *its interaction with surrounding objects*. ($ii$) **Referring Reasoning**: models need to reason on top of one or more referred regions correctly. ($iii$) **Grounding in Conversation**: models are required to reason correctly and accurately ground/localize the objects/regions necessary for the reasoning. For the ease of benchmarking other methods, we represent the regions with boxes instead of points or free-form shapes.

Specifically, we randomly sample 40 images from the COCO validation set for each type of question, and generate the questions and GPT-4's answers following the instruction generation pipeline in Sec. 3.2. Following Liu et al. (2023b), we feed the question and image into MLLMs to obtain the predicted answer, and then prompt GPT-4 to rate the predicted answer and pseudo answer from GPT-4 based on the ground-truth textual scene description (object, relationship, region caption, global caption). GPT-4 evaluates both the precision of referring understanding, object grounding, and correctness of semantics. The rating score ranges from 1 to 10, in which higher means better. We calculate the ratio of the predicted answer's score and the GPT-4 answer's score, which is then presented as a percentage to measure the performance of MLLMs. We also asked GPT-4 to give a comprehensive review for the rating and found that GPT-4 is good at measuring the degree of spatial precision, such as how much the predicted bounding box diverges from the GT box coordinate. We refer the readers to Appendix D for further elaboration.

Table 4: Results on LLaVA-Bench and the proposed Ferret-Bench via GPT4-as-a-Judge evaluation.

| | LLaVA-Bench | | | | Ferret-Bench | | | |
|---|---|---|---|---|---|---|---|---|
| | Conversation | Detail Description | Complex Reasoning | Avg. | Referring Description | Referring Reasoning | Grounding in Conversation | Avg. |
| LLaVA[8] | 85.4 | 68.3 | 92.1 | 81.9 | 41.4 | 31.7 | 28.8 | 34.0 |
| Kosmos-2 | 71.7 | 63.4 | 74.9 | 70.0 | 51.8 | 33.7 | 48.4 | 44.6 |
| Shikra-7B | 80.6 | 70.7 | 88.1 | 79.9 | 46.0 | 41.6 | 50.1 | 45.9 |
| Ferret-7B | 84.4 | 79.4 | 96.3 | 86.7 | 68.7 | 67.3 | 57.5 | 64.5 |
| Ferret-13B | **85.2** | **80.9** | **96.4** | **87.5** | **70.6** | **68.7** | **59.7** | **66.3** |

Table 5: Ablation study on the mutual benefit of grounding data and referring data. We evaluate Accuracy for LVIS referring and R@1 for grounding.

| Model | Referring (LVIS) | | Grounding |
|---|---|---|---|
| | Point | Box | Flickr30k |
| Ferret | **67.9** | **79.4** | **80.4** |
| w/o Grounding data | 65.4 | 75.6 | ✗ |
| w/o Referring data | ✗ | ✗ | 79.8 |

Table 6: Ablation study on the effectiveness of the proposed spatial-aware visual sampler. Accuracy is used to evaluate LVIS referring.

| Module | Referring (LVIS) | | |
|---|---|---|---|
| | Point | Box | Free-form |
| Spatial-aware Visual Sampler | **67.9** | **79.4** | **69.8** |
| Visual Sampler in SEEM | 67.1 | 77.2 | 68.9 |

We use LLaVA-Bench (Liu et al., 2023b) and the proposed Ferret-Bench to compare Ferret with previous models, including LLaVA (Liu et al., 2023b), Shikra (Chen et al., 2023b), and Kosmos-2 (Peng et al., 2023). Results are summarized in Table 4. Ferret achieves superior performance in all types of tasks, boosting the score for the detailed description category from 68.3 to 80.9, and especially excels at the three new tasks demanding referring and grounding abilities.

## 4.4 ABLATION

In the ablation studies below, in default, we ablate Ferret-7B and mainly evaluate in referring object classification and grounding tasks on Flickr30k Entities validation set.

**Mutual benefits of grounding and referring.** As shown in Table 5, grounding and referring, as two main capabilities emphasized in this paper, can actually benefit each other. Particularly, when adding grounding data into training, the referring performance gets improved, and vice versa.

**Spatial-aware Visual Sampler.** We ablate the effectiveness of the spatial-aware visual sampler by replacing it with the visual sampler in SEEM (Zou et al., 2023), where they average the features of all the sampled points as the region feature. As we can see in Table 6, ours can outperform the previous visual sampler in all three referring tasks.

**LLM model size.** We study how much LLM model size influences the performance of referring and grounding. As seen in Table 1-4, having a larger LM backbone can generally help.

## 4.5 OBJECT HALLUCINATION

Attribute to the incorporation of fine-grained spatial knowledge and negative mining, Ferret also exhibits strong power against the hallucination problem. We evaluate object hallucinations on the POPE benchmark (Li et al., 2023e). Results are summarized in Table 7. Ferret has exhibited performance comparable to Shikra (Chen et al., 2023b), and far surpasses recent popular MLLMs.[9]

## 5 CONCLUSION

We present Ferret, a new multimodal large language model adept at referring and grounding. Ferret can refer image regions in any free-form shape, and automatically establish grounding for text deemed groundable by the model. We have curated the GRIT dataset for model training, and the Ferret-Bench dataset for evaluation. Ferret, like most MLLMs, may produce harmful and counterfactual responses. For future work, inspired by LISA (Lai et al., 2023), we plan to enhance Ferret to be able to output segmentation masks in addition to bounding boxes.

---

[8]The result on LLaVA-Bench is obtained by evaluating LLaVA released checkpoint. The slight discrepancy might be due to evolving GPT4 APIs. For Ferret-Bench, we employ the same conversation template as Ferret, providing LLaVA with a predefined input size, resizing all coordinates accordingly, and generating a response.

[9]Unlike other methods, Ferret refrains from relying on VQA. This decision stems from our observation that VQA answers tend to be concise, and this brevity can restrict the conversational capabilities of LLMs.

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

Table 7: Results on the object hallucination benchmark using the POPE evaluation pipeline (Li et al., 2023e).

| Datasets | Metrics | Ferret | Shikra | InstructBLIP | MiniGPT4 | LLaVA | MM-GPT | mPLUG-Owl |
|---|---|---|---|---|---|---|---|---|
| Random | Accuracy (↑) | **90.24** | 86.90 | 88.57 | 79.67 | 50.37 | 50.10 | 53.97 |
| | Precision (↑) | 97.72 | 94.40 | 84.09 | 78.24 | 50.19 | 50.05 | 52.07 |
| | Recall (↑) | 83.00 | 79.26 | 95.13 | 82.20 | 99.13 | 100.00 | 99.60 |
| | F1 Score (↑) | 89.76 | 86.19 | 89.27 | 80.17 | 66.64 | 66.71 | 68.39 |
| | Yes | 43.78 | 43.26 | 56.57 | 52.53 | 98.77 | 99.90 | 95.63 |
| Popular | Accuracy (↑) | **84.90** | 83.97 | 82.77 | 69.73 | 49.87 | 50.00 | 50.90 |
| | Precision (↑) | 88.24 | 87.55 | 76.27 | 65.86 | 49.93 | 50.00 | 50.46 |
| | Recall (↑) | 80.53 | 79.20 | 95.13 | 81.93 | 99.27 | 100.00 | 99.40 |
| | F1 Score (↑) | 84.21 | 83.16 | 84.66 | 73.02 | 66.44 | 66.67 | 66.94 |
| | Yes | 45.63 | 45.23 | 62.37 | 62.20 | 99.40 | 100.00 | 98.57 |
| Adversarial | Accuracy (↑) | 82.36 | **83.10** | 72.10 | 65.17 | 49.70 | 50.00 | 50.67 |
| | Precision (↑) | 83.60 | 85.60 | 65.13 | 61.19 | 49.85 | 50.00 | 50.34 |
| | Recall (↑) | 80.53 | 79.60 | 95.13 | 82.93 | 99.07 | 100.00 | 99.33 |
| | F1 Score (↑) | 82.00 | 82.49 | 77.32 | 70.42 | 66.32 | 66.67 | 66.82 |
| | Yes | 48.18 | 46.50 | 73.03 | 67.77 | 99.37 | 100.00 | 98.67 |

Table 8: Comparison of Ferret *v.s.* recent MLLMs integrating spatial awareness. 'Convention' refers to a comprehensive collection of publicly available data that has been transformed using templates, 'GPT-Generate' signifies the generated refer/ground datasets employing GPT, and 'Robustness' denotes datasets aimed at mitigating hallucination and improving robustness. Section 3 explains more details about each.

| Model | Input Types | | | Output Grounding | Data Construction | | | Quantatitive Eval. of Refer/Ground w. Chat |
|---|---|---|---|---|---|---|---|---|
| | Point | Box | Free-form | | Convention | GPT-Generate | Robustness | |
| BuboGPT | ✗ | ✗ | ✗ | ✔ | ✔ | ✗ | ✗ | ✗ |
| Vision-LLM | ✗ | ✗ | ✗ | ✔ | ✔ | ✗ | ✗ | ✗ |
| Kosmos-2 | ✗ | ✔ | ✗ | ✔ | ✔ | ✗ | ✗ | ✗ |
| Shikra | ✔ | ✔ | ✗ | ✔ | ✔ | ✔ | ✗ | ✗ |
| GPT4-ROI | ✗ | ✔ | ✗ | ✗ | ✔ | ✗ | ✗ | ✗ |
| PVIT | ✗ | ✔ | ✗ | ✗ | ✔ | ✔ | ✗ | ✔ |
| **Ferret** | ✔ | ✔ | ✔ | ✔ | ✔ | ✔ | ✔ | ✔ |

# A   RELATED WORK

**Multimodal large language models (MLLMs).**   Large Language Models (LLMs), including GPTs (Brown et al., 2020; OpenAI, 2023a), PaLM (Chowdhery et al., 2022), BLOOM (Scao et al., 2022), and LLaMA (Touvron et al., 2023a;b), have revolutionized research in NLP, spurring significant advances in multimodal language models as well. Early models primarily focused on large-scale image-text pre-training. Notable examples include SimVLM (Wang et al., 2022c), GIT (Wang et al., 2022a), PaLI (Chen et al., 2022b), PaLI-X (Chen et al., 2023c), BLIP-2 (Li et al., 2023c), Flamingo (Alayrac et al., 2022), PaLM-E (Driess et al., 2023), CM3 (Aghajanyan et al., 2022), and CM3Leon (Yu et al., 2023). Flamingo, in particular, pioneered the integration of a pre-trained CLIP image encoder with LLMs through gated cross-attention blocks, showcasing emergent multimodal in-context few-shot learning capabilities. Its open-sourced variants, such as OpenFlamingo (Awadalla et al., 2023) and IDEFICS (Laurençon et al., 2023), have garnered significant attention. Typically, these models undergo pre-training using millions or even billions of image-text pairs and interleaved image-text datasets (Zhu et al., 2023b).

On the other hand, recent research has increasingly focused on using pre-trained LLMs for visual instruction tuning. Prominent examples include LLaVA (Liu et al., 2023b), MiniGPT-4 (Zhu et al., 2023a), mPLUG-Owl (Ye et al., 2023), Otter (Li et al., 2023a), InstructBLIP (Dai et al., 2023), to name a few. In addition to text generation, recent models like FROMAGe (Koh et al., 2023b), GILL (Koh et al., 2023a), Emu (Sun et al., 2023), have also enabled MLLMs for image retrieval and image generation. Please refer to Chapter 5 of Li et al. (2023b) for a detailed review.

**MLLMs for referring and grounding.** In the realm of existing literature, works such as Kosmos-2 (Peng et al., 2023) and Shikra (Chen et al., 2023b), closely resemble ours as they also enable MLLMs for fine-grained image comprehension and open-world referring and grounding. Additional works in this direction include GPT4ROI (Zhang et al., 2023), PVIT (Chen et al., 2023a), BuboGPT (Zhao et al., 2023), VisionLLM (Wang et al., 2023), and ContextDET (Zang et al., 2023). Nevertheless, pivotal distinctions set our model apart. First, prior endeavors supported only bounding boxes (and points in Shikra) as input. Conversely, due to Ferret's innovative hybrid region representation, we accommodate a broader range of free-form shapes for referring, encompassing points, boxes, sketches, scribbles, polygons, and more. Second, we meticulously curate an extensive refer-and-ground instruction tuning dataset. Third, we introduce Ferret-Bench to facilitate forthcoming research and enhance evaluation benchmarks in this direction. Lastly, our model exhibits superior performance compared to previous works, notably mitigating object hallucination to a significant extent. A more straightforward side-by-side comparison is shown in Tab. 8.

**Unifying grounding and VL understanding.** Our work is also related to previous work that aims to unify text and bounding box output for vision-language (VL) models, such as UniTAB (Yang et al., 2022), OFA (Wang et al., 2022b), and Unified-IO (Lu et al., 2022), which also represent bounding boxes using a set of additional discrete tokens as proposed in Pix2Seq (Chen et al., 2021; 2022a). Ferret is unique in that ($i$) our model is built upon LLMs, marrying the power of LLMs and grounding, thus unlocking new capabilities such as grounded instruction tuning, and ($ii$) we handle bounding box coordinates as regular text tokens, avoiding the need for extra specialized tokens dedicated to representing boxes.

## B    Discussion on Limitation and Failure Cases

We acknowledge certain specific failure scenarios and limitations for our models, which are detailed as follows:

**Failure Scenarios**: (1). Referring to too many objects (more than 3) in one question might not be as accurate as referring to each of them in separate conversations. This is likely due to a relative scarcity of training data that mentions too many objects. (2). The referring and grounding of very small objects is less accurate than large or medium objects. It's a common challenge in object detection. However, we think further improving input image resolution is able to help.

**Limitations**: (1). Not good at other languages because the training dataset is curated only in English. Although Ferret shows some emergent referring and grounding capability in other languages, its performance in other languages is still worse than in English. Future incorporation of multilingual training data could potentially mitigate this. (2). Similar to many large language models, Ferret has the potential to generate harmful or factually incorrect responses. (3). Ferret is not designed for segmentation tasks requiring mask outputs.

## C    Details of Dataset

### C.1    Task Templates for Public Datasets

In Section 3.1, we mentioned using carefully designed task templates to convert public datasets such as Visual Genome into instruction-following format. The task templates we used are provided in Table 9. For simplicity, we only list three examples for each task.

### C.2    Details on Spatial Negative Mining

In Section 3.3, we conducted negative sample mining for two aspects: ($i$) *Image-conditioned Category Localization*, and ($ii$) *Semantics-conditioned Category Localization*. They use the same template to convert the original data, which falls into the task of object hallucination in Table 9. Specifically, for the negative categories in ($ii$), we prompt ChatGPT/GPT-4 to generate entities that are most analogous to the original class, attribute, or quantity, *e.g.*, 'man' vs. 'woman', 'blue' vs. 'yellow', 'two' vs. 'three'. The prompt feed into ChatGPT/GPT-4 encompasses all the entities extracted from 5 captions associated with one single image. We show the exact prompt template in Table 10.

Table 9: Examples of task templates Ferret used to transfer different public data types into the instruction-following format.

| Task | Three randomly chosen examples from many. |
|---|---|
| Referring-Object | What is the class of the object <location> within the image?
Classify object <location> in the image.
Identify the object <location> in the image. |
| Referring-Relation | What does <object1> <location1> do to <object2> <location2> of the image?
What is the physical relation between <object1> <location1> and <object2> <location2>?
Can you figure out the geometric relation of the <object1> <location1> and <object2> <location2>? |
| Referring-Region | Describe the region <location> in a short phrase.
What is in the region <location>? Describe in a phrase.
Capture in a phrase: what's near region <location> in the picture? |
| REC. | Where is <object> in the image?
What are the coordinates for the given <object> in the image?
Given the image, could you please tell me where is <object> |
| Phrase Grounding | What are the locations of <objects>?
Could you provide me with the exact locations of <objects>?
Please indicate the positions of <objects> in the image? |
| Object Detection (O365) | Detect all objects among <class> in the image.
Perform object detection given the image within <class>.
Given the image and set <class>, identify all the objects that belong to the set. |
| Grounded Captioning | What is this photo about? Use concise language.
Describe the overall picture in just a few words.
What do you see happening in this image? Provide the answer in short. |
| Object Hallucination | Is there a <object> in the image?
Are there <object> in the image?
Please tell me whether <object> exists in the image? |

Table 10: In this example, we provide the prompt to generate the spatial negative sets.

messages = [ {"role":"system", "content": f'''You are an AI visual assistant that can analyze a single image. You receive **several entities** given by a list, each describing the objects in the image you are observing.

For each entity mentioned, change them with the most misleading entity name (may belong to the same category but are actually different) (**nonexistent objects**: man → woman, **nonexistent attributes**: brown → yellow, **nonexistent quantities**: two → three, etc.). The instructions should contain interrogative and declarative sentences.

The output format needs to be a list only which contains the misleading entity names. Please follow the instructions carefully.

1. The length of the output list needs to be exactly equal to the input list.

2. Do not explain the reasons.

3. Do not mention the input entities, at least the output name and input name needs to be different.

4. Do not mention something abstract, like älien.

5. When dealing with quantities, focus solely on increasing the numbers during revision.

6. When dealing with words like "a few", "a group", "several", "some", etc., try changing the objects (A few men → A few women).

7. Ensure that inclusive words are not substituted with their specific subsets. For example, if the word is "people," avoid replacing it with genders like "man" or "woman." Instead, consider modifying them to different categories, such as "people" → "animals.".'''}]

## C.3 EXAMPLES FOR GENERATING REFER-AND-GROUND DATASETS

We provide some example prompts to generate refer-and-ground from ChatGPT/GPT-4. Prompt and the in-context example of multiple-round visual conversation data are shown in Table 11 and Table 12. Prompt and the in-context example of one-round reasoning data are shown in Table 13 and Table 14.

Table 11: In this example, we provide the prompt used to generate the conversation response for refer-and-ground instruction tuning, following the practice of LLaVA (Liu et al., 2023b).

```
messages = [ {"role":"system", "content": f"'You are an AI visual assistant that
can analyze a single image. You receive five global captions, each describing the same image you
are observing. In addition, specific object locations within the image are given, along with detailed
coordinates. These coordinates are in the form of bounding boxes, represented as (x1, y1, x2, y2)
with floating numbers ranging from 0 to 1. These values correspond to the top left x, top left y,
bottom right x, and bottom right y. Also, the relationships between pairs of objects are provided in
the format of object → relationship → subject, where the object/subject are indexed by object id
from previous object lists as well as the object names. Also, several region descriptions are given,
each describing a box region of the image, with detailed coordinates.

Design a conversation between you and a person asking about this photo. Ask diverse questions and
give corresponding answers. The answers should be in a tone that a visual AI assistant is seeing the
image and answering the question.

Here are some additional requirements about generated questions and answers:

1. Only include questions that have definite answers:
(1) one can see the content in the image that the question asks about and can answer confidently;
(2) one can determine confidently from the image that it is not in the image. Do not ask any
questions that cannot be answered confidently.

2. Also include complex questions that are relevant to the content in the image, for example, asking
about background knowledge of the objects in the image, asking to discuss events happening in the
image, asking about object actions in the context of entire images, etc. Again, do not ask about
uncertain details.

3. Provide detailed answers when answering complex questions. For example, give detailed
examples or reasoning steps to make the content more convincing and well-organized. You can
include multiple paragraphs if necessary.

4. In all samples, either in question or answer, you must mention bounding box coordinates to refer
to the object or regions instead of directly saying the object name or describing the regions in text.
In answer, explain the region in the context of the scene.

5. Do not mention that the information source is provided in the text/caption/region description.
Always answer as if you are directly looking at the image.

6. Make the question as diverse as possible. Include questions asking about the visual content of
the image, including the object types, counting the objects, object actions, object locations, relative
positions between objects, object selection, object functions, etc. Make the question challenging by
less including the visual content details in the question."'}
]
for sample in fewshot_samples:
    messages.append({"role":"user", "content":sample['context']})
    messages.append({"role":"assistant", "content":sample['response']}
    )
messages.append({"role":"user", "content":'\n'.join(query)})
```

Table 12: One example used in in-context learning to construct GPT-Assisted Refer-and-Ground Instructon-Tuning. We encourage readers to refer to the codebase for the prompts for other types.

**Objects**
Object 0: chair at [0.596, 0.637, 0.698, 0.997].
...
**Relationships**
Object 11 : frame → with → Object 9 : photo
...
**Region Descriptions**
Region Description at [0.560, 0.466, 0.600, 0.529] :
        a white picture frame with a black and white photo on it.
...
**Global Caption**
White chairs sit around a polished wood dining table while
        a sectional soft sits in the background.
...

**Conversation**
Question:
What is the object [0.212, 0.539, 0.718, 0.820] made of?
===
Answer:
The dining table over there is crafted from a tree trunk.
===
Question:
Where would be a good place to have my dinner? Point out the locations in image.
===
Answer:
I suggest having dinner at the table [0.214, 0.541, 0.720, 0.997], possibly sitting on the white chair [0.264, 0.717, 0.524, 0.995].
Also, enjoy the wine [0.242, 0.644, 0.288, 0.682].
===
Question:
In terms of comfort for resting, which object is better - the one [0.830, 0.431, 0.978, 0.596] or the one [0.264, 0.717, 0.524, 0.995]?
===
Answer:
Among the two, the couch [0.830, 0.431, 0.978, 0.596] would be more comfortable for resting. The object [0.264, 0.717, 0.524, 0.995] is a dining chair, which is typically less comfortable for extended relaxation.
===
Question:
Is the item [0.182, 0.238, 0.342, 0.461] real or is it a mirror reflection?
===
Answer:
The television and the fireplace [0.182, 0.238, 0.342, 0.461] are not real but are actually reflections in the mirror.
===
Question:
Can you describe the decorating style of the living room shown in this image?
===
Answer:
The living room is decorated in a modern chic style. The image shows white chairs and lamps, a gray couch, and a wooden table and floor, all contributing to a modern aesthetic. The large mirror and built-in wine rack are also characteristic of modern design. Thus, the overall decor creates an atmosphere of elegance and sophistication, indicating a modern chic style.

Table 13: In this example, we provide the prompt used to generate the reasoning response for refer-and-ground instruction tuning, following the practice of LLaVA (Liu et al., 2023b).

```
messages = [ {"role":"system", "content": f'''You are an AI visual assistant that
can analyze a single image. You receive five global captions, each describing the same image you
are observing. In addition, specific object locations within the image are given, along with detailed
coordinates. These coordinates are in the form of bounding boxes, represented as (x1, y1, x2, y2)
with floating numbers ranging from 0 to 1. These values correspond to the top left x, top left y,
bottom right x, and bottom right y. Also, the relationships between pairs of objects are provided, in
the format of object → relationship → subject, where the object/subject are indexed by object id
from previous object lists as well as the object names. Also, several region descriptions are given,
each describing a box region of the image, with detailed coordinates.

The task is to use the provided image information (objects, attribute, relationship, region description,
captions), create a plausible and challenging question about the image, and provide the answer in
detail.

Create complex questions that mention specific regions of the image, but the question should require
some knowledge-aware or high-level commonsense reasoning beyond describing the scene.

To answer such questions, one should first understand the visual content, then based on the
background knowledge or reasoning, either explain why the things are happening that way or
provide guides and help to the user's request. Make the question challenging by not including the
visual content details in the question so that the user needs to reason about that first.

Here are some additional requirements about generated questions and answers:

1. In question or answer, you must mention bounding box coordinates to refer to the object or
regions, instead of directly say the object name or describing the regions in text. In answers, explain
the region in the context of scene. Include details like object counts, position of the objects, relative
position between the objects.

2. Don't ask the question you are not confident to answer. Only include question that have definite
answer.

3. Do not mention that the information source is provided in text/catpion/region description. Always
answer as if you are directly looking at the image.

4. Make the question as diverse as possible and as complex-reasoning required as possible.'''}
]
for sample in fewshot_samples:
    messages.append({"role":"user", "content":sample['context']})
    messages.append({"role":"assistant", "content":sample['response']}
    )
messages.append({"role":"user", "content":'\n'.join(query)})
```

Table 14: One example used in in-context learning to construct GPT-Assisted Refer-and-Ground Instructon-Tuning. We encourage readers to refer to the codebase for the prompts for other types.

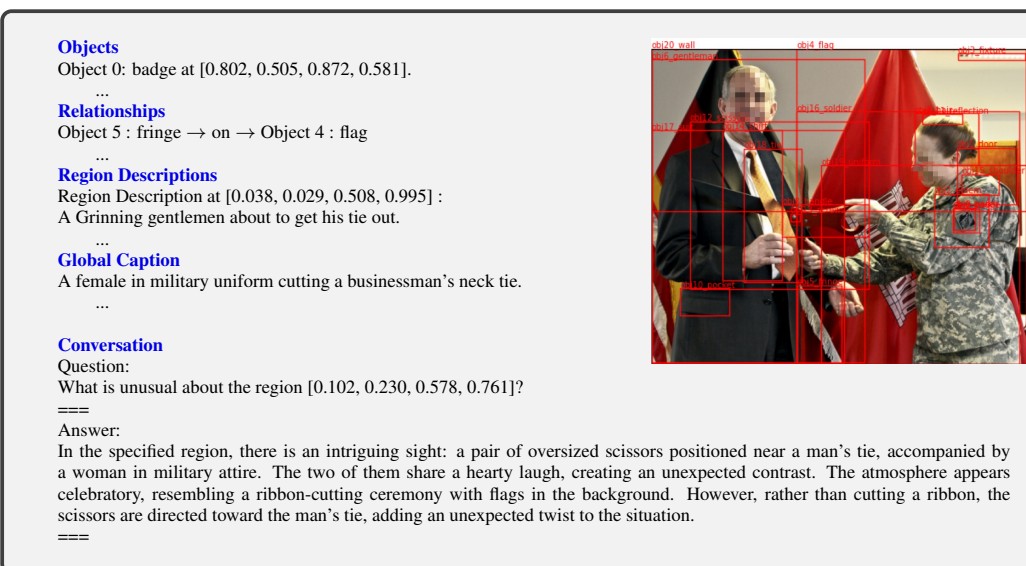

**Objects**
Object 0: badge at [0.802, 0.505, 0.872, 0.581].
...
**Relationships**
Object 5 : fringe → on → Object 4 : flag
...
**Region Descriptions**
Region Description at [0.038, 0.029, 0.508, 0.995] :
A Grinning gentlemen about to get his tie out.
...
**Global Caption**
A female in military uniform cutting a businessman's neck tie.
...

**Conversation**
Question:
What is unusual about the region [0.102, 0.230, 0.578, 0.761]?
===
Answer:
In the specified region, there is an intriguing sight: a pair of oversized scissors positioned near a man's tie, accompanied by a woman in military attire. The two of them share a hearty laugh, creating an unexpected contrast. The atmosphere appears celebratory, resembling a ribbon-cutting ceremony with flags in the background. However, rather than cutting a ribbon, the scissors are directed toward the man's tie, adding an unexpected twist to the situation.
===

# D   EXAMPLES AND PROMPTS FOR FERRET-BENCH

We leverage GPT-4 to generate three kinds of region-based questions evaluating referring and grounding capability: (i) Referring Description, (ii) Referring Reasoning, and (iii) Grounding in Conversation. Here, we only provide the prompt in Table 15 used to generate the referring description response. One example of GPT-4 answers is shown in Table 16. We recommend readers check out more examples in Appendix E.

Table 15: In this example, we provide the prompt used to generate the referring description response.

```python
messages = [ {"role":"system", "content": f'''You are an AI visual assistant that
can analyze a single image. You receive five global captions, each describing the same image you
are observing. In addition, specific object locations within the image are given, along with detailed
coordinates. These coordinates are in the form of bounding boxes, represented as (x1, y1, x2, y2)
with floating numbers ranging from 0 to 1. These values correspond to the top left x, top left y,
bottom right x, and bottom right y. Also, the relationships between pairs of objects are provided, in
the format of object → relationship → subject, where the object/subject are indexed by object id
from previous object lists as well as the object names. Also, several region description are given,
each describing a box region of image, with detailed coordinates.

The task is to use the provided image information (objects, attribute, relationship, region description,
captions), create a plausible and challenging question about the image, and provide the answer in
detail.

Create questions that refer to coordinates of some objects or regions without describing it, and ask
about its interaction with surrounding/nearby objects.

To answer such questions, one should require first understanding the visual content, then based on
the spatial information provided.

Here are some additional requirements about generated questions and answers:

1. In question, you must mention bounding box coordinates to refer to the object or regions, instead
of directly say the object name or describing the regions in text. In answers, explain the region in
the context of scene. Include details like object counts, position of the objects, relative position
between the objects.

2. Don't ask the question you are not confident to answer. Only include question that have definite
answer.

3. Do not mention that the information source is provided in text/catpion/region description. Always
answer as if you are directly looking at the image.

4. Don't mention additional coordinates in the answer.

5. Question should be explicitly ask about context/surrounding/nearby information/interaction.'''}
]
for sample in fewshot_samples:
    messages.append({"role":"user", "content":sample['context']})
    messages.append({"role":"assistant", "content":sample['response']}
    )
messages.append({"role":"user", "content":'\n'.join(query)})
```

Table 16: One example of GPT4's answers about referring description.

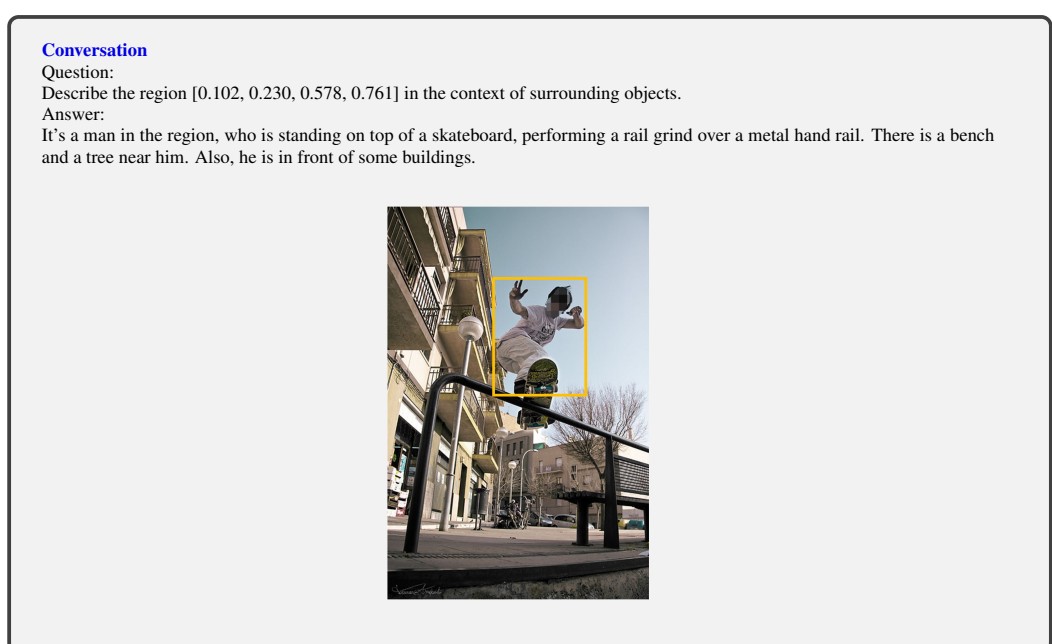

# E  MORE VISUALIZATION

We provide more quantitative results of the predictions under various tasks from Ferret to indicate the model's strength and capability.

- Please refer to Figure 5 for Referring Object Classification on LVIS with different referring formats (point/box/).
- Please refer to Figure 6 for Visual Grounding on Flickr30k Entities and Referring Expression Comprehension on RefCOCO/RefCOCO+/RefCOCOg.
- Please refer to Figure 7 for Grounded Captioning on Flickr30k Karpathy split.
- Please refer to Figure 8 for Evaluating Object Hallucination (POPE) on COCO val split.
- Please refer to Table 17 for Referring Description in Ferret-Bench.
- Please refer to Table 18 for Referring Resoning in Ferret-Bench.
- Please refer to Table 19 for Grounding in Conversation in Ferret-Bench.

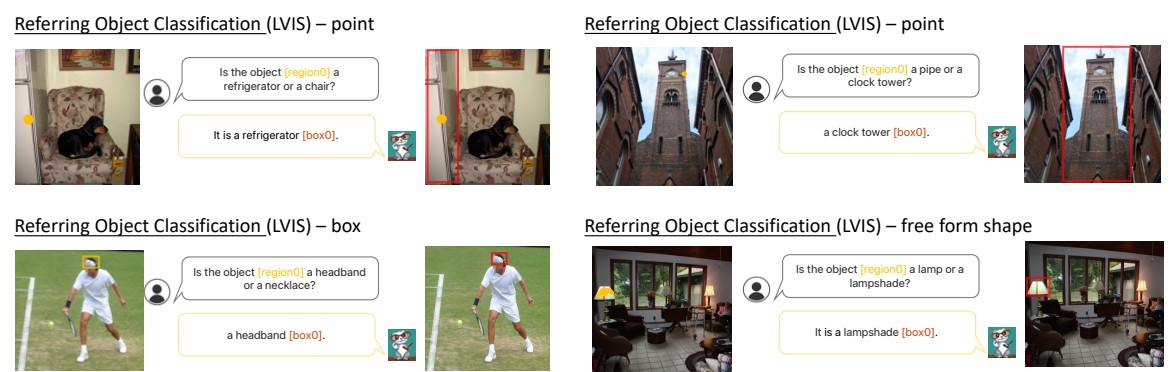

Figure 5: **Referring Object Classification on LVIS**. The task aims to classify specific region(s) in an image given by point/box/segmentation inputs.

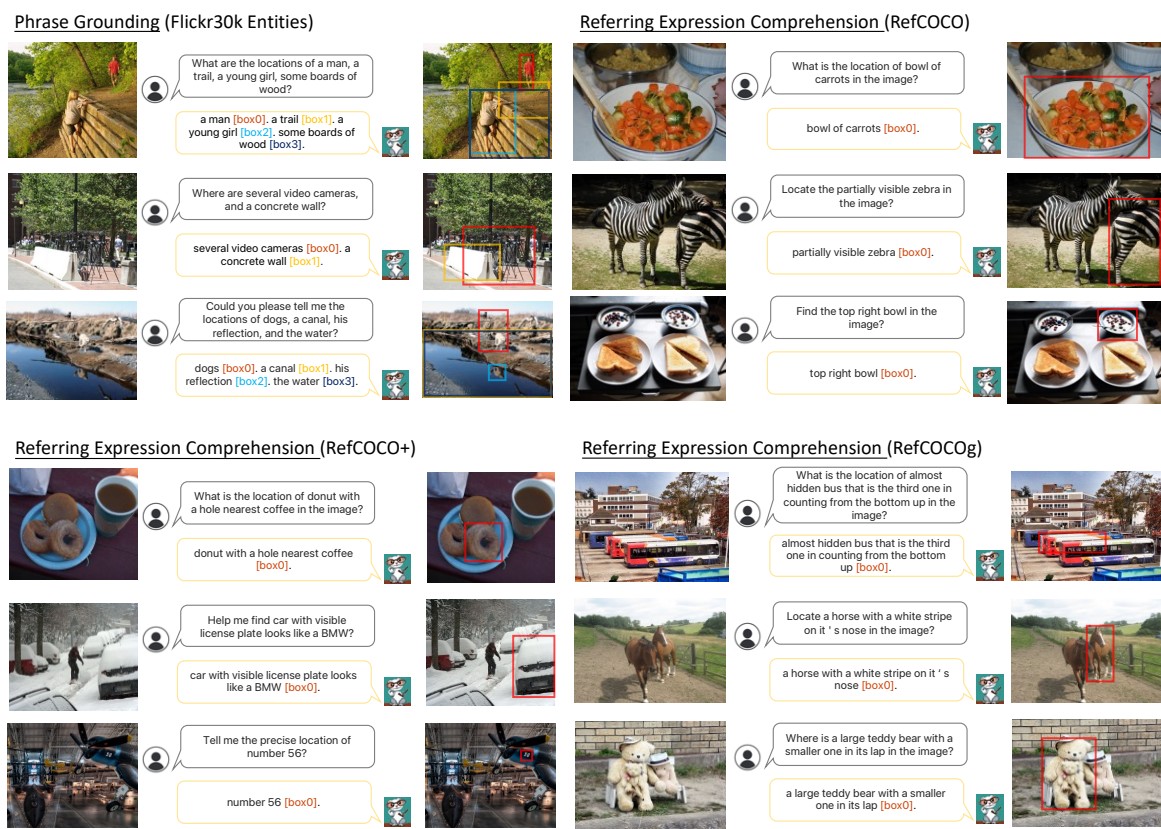

Figure 6: **Phrase Grounding** on Flickr30k Entities and **Referring Expression Comprehension** on RefCOCO/RefCOCO+/RefCOCOg. The tasks aim to localize specific object(s) in an image described by a referring expression/given entity.

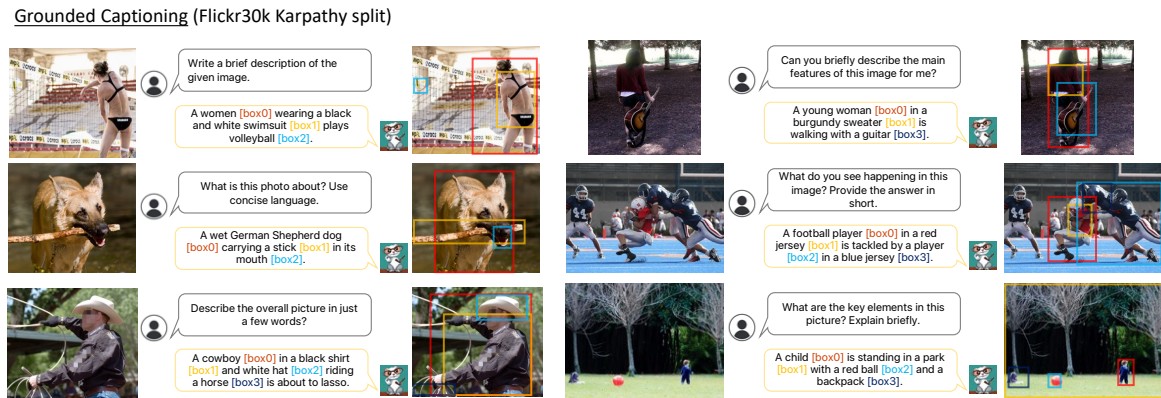

Figure 7: **Grounded Captioning on Flickr30k**. The task aims to generate a caption about the image and ground all generated noun phrases to image regions.

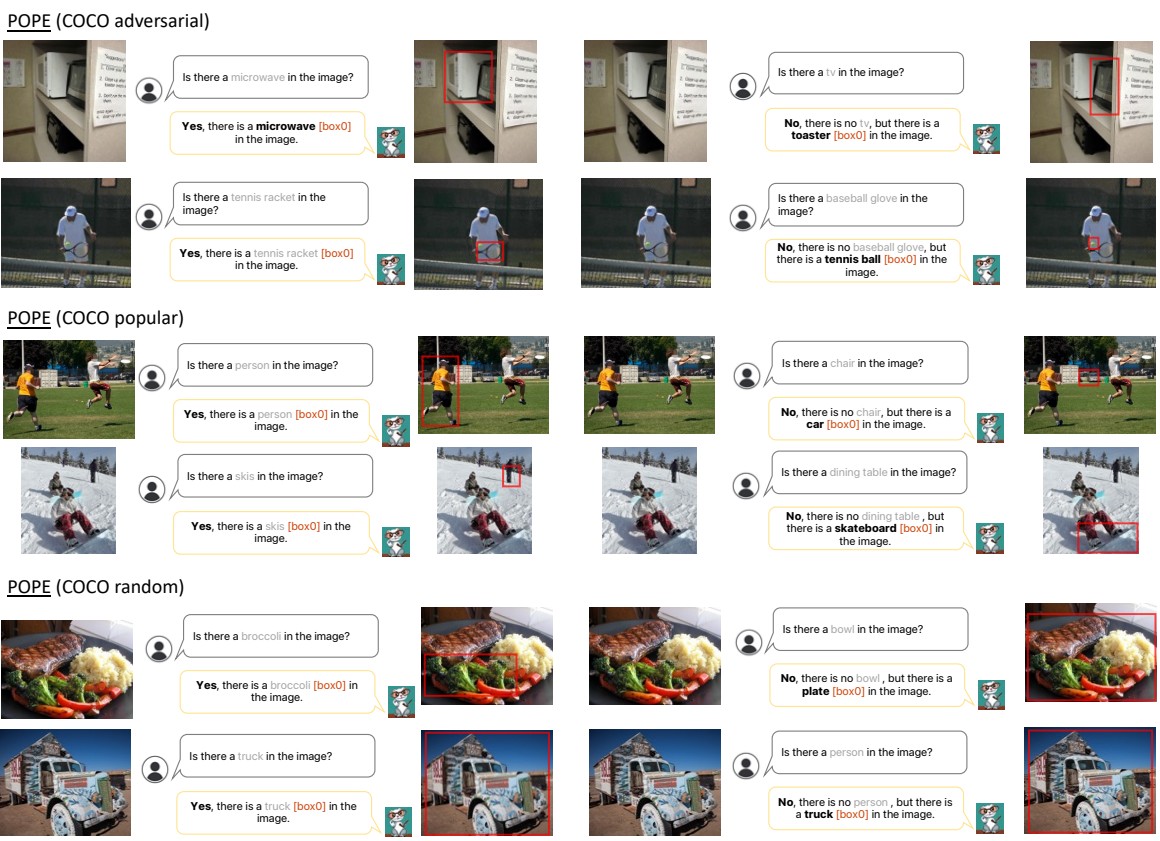

Figure 8: **Object Hallucination Evaluation (POPE) on COCO**. The task aims to input a query inquiring about the existence of an object, and the model is expected to generate a response in the form of either "yes/no".

Table 17: **Referring Description in Ferret-Bench.** Qualitative examples to illustrate the difference between various models (LLaVA vs. Kosmos-2 vs. Shikra vs. Ferret (Ours)). For clarity, we have omitted the bounding box outputs in the textual representations.

**Question:**
What is the interaction between the object [region0] and its surrounding objects?

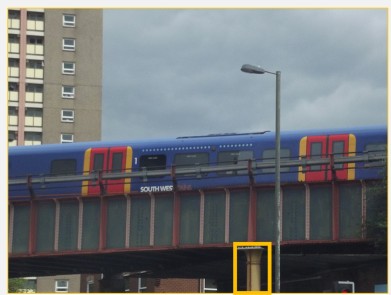

**Ground Truth:**
In the region, there is a metal support column. This column is providing support for a bridge above it, which a train is passing over. The column also features a red line on it. This region is part of a larger scene that includes a train track on an elevated bridge.

**LLaVA:**
The object is a bridge which appears to be a part of a train station, as there are traffic lights present in the scene. The traffic lights are likely used to control the train's movement and ensure safe passage over the bridge.

**Kosmos-2:**
The interaction between the object and its surroundings is that of a train passing over a bridge.

**Shikra:**
The metal pole is standing tall and straight, supporting the train that is traveling on the elevated track.

**Ferret (Ours):**
The object is a metal post that is part of a bridge. This post is essential because it supports the bridge structure, ensuring its stability for the train to safely traverse the bridge. Other elements such as the train, and the overhead guard rail, depend on this post for support and stability. Without this post, the bridge wouldn't be able to withstand the train's weight and motion, potentially leading to a structural collapse. Therefore, this post plays a critical role in maintaining the bridge's integrity and stability.

---

**Question:**
What is the relationship between the object [region0] and object [region1] and its surrounding objects?

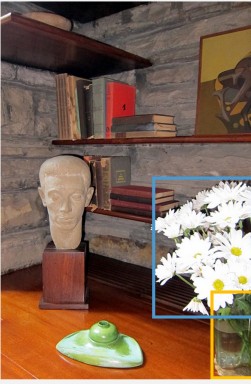

**Ground Truth:**
The object is a vase, and the object is a flower. The flower is in the vase, suggesting it is a decorative element within the room.

**LLaVA:**
A wooden table with a green bowl placed on it.

**Kosmos-2:**
The object is a vase with a green base.

**Shikra:**
The answer is flowers.

**Ferret (Ours):**
The object and object are a vase and a flower, respectively. The vase is holding the flower, suggesting it's displaying or supporting it.

Table 18: **Referring Reasoning in Ferret-Bench.** Qualitative examples to illustrate the difference between various models (LLaVA vs. Kosmos-2 vs. Shikra vs. Ferret (Ours)). For clarity, we have omitted the bounding box outputs in the textual representations.

**Question:**
What is the purpose of the object [region0]?

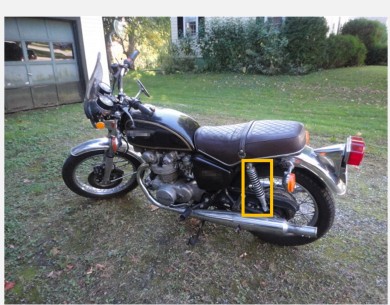

**Ground Truth:**
The object is a bottle. Bottles are typically used to hold and store different types of liquids. In this context, it might be used to store a beverage for the meal.

**LLaVA:**
A knife and a fork, indicating that it is ready for a meal.
**Kosmos-2:**
The purpose of the object is to attract the birds to the table.
**Shikra:**
The answer is keep birds away.
**Ferret (Ours):**
The object is a bottle, generally used to store and dispense liquids like water, juice, or other beverages.

---

**Question:**
What is the purpose of the object [region0] on the bike?

**Ground Truth:**
The object is a shock absorber on the bike. Its purpose is to absorb or dampen shock impulses. It does this by converting the kinetic energy of the shock into another form of energy which is then dissipated. In the context of the motorcycle, it is particularly useful in providing comfort and stability for the rider, especially when travelling over uneven or rough terrains.

**LLaVA:**
A black and silver Kawasaki motorcycle which is likely used for transportation, commuting, leisure rides, or long-distance travel.
**Kosmos-2:**
The purpose of the object on the motorcycle is to store and protect the motorcycle's fuel tank.
**Shikra:**
The answer is leg.
**Ferret (Ours):**
The object is a shock absorber. This component is essential for the bike's stability, providing a smoother ride by soaking up the impact from bumps and uneven surfaces. It significantly improves the overall performance and comfort of the bike, enabling the rider to maintain control and stability even on rough terrains.

Table 19: **Grounding in Conversation in Ferret-Bench.** Qualitative examples to illustrate the difference between various models (Kosmos-2 vs. Shikra vs. Ferret (Ours)). For clarity, we only display the bounding box outputs in Ferret 's responses.

**Question:**
What is happening in the field and
provide the coordinates of mentioned objects?

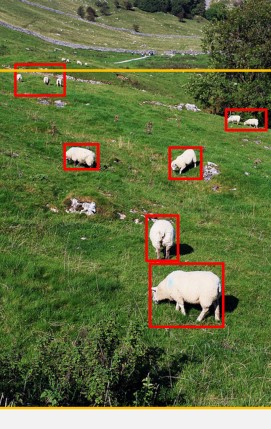

**Ground Truth:**
In the image, there is a group of sheep [box0] [box1]
grazing in a field [box2]. The field is filled with
plant life [box3]. There is also a bush [box4] and
some trees [box5] present in the field.

**Kosmos-2:**
In the field, a herd of sheep [box0] is grazing on the
lush green grass [box1].
**Shikra:**
A group of sheep [box0] [box1] [box2] [box3] [box4] [box5]
grazing on lush green grass.
**Ferret (Ours):**
In the field, there are sheep [box0] [box1] [box2] [box3] [box4] [box5] grazing on the grass [box6].

---

**Question:**
What is the man doing and what objects are in front of him?
Please provide the coordinates of these objects.

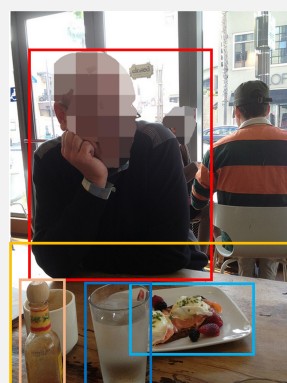

**Ground Truth:**
The man [box0] is sitting at a table [box1] and looking
at a plate of food [box2] . There is a glass [box3]
and a bottle [box4] on the table in front of him.

**Kosmos-2:**
The man is sitting at a dining table [box0] smiling and
looking at the plate of food in front him [box1].
**Shikra:**
The man [box0] is sitting at a table [box1] with a plate
of food [box2] in front of him. He is looking up and
smiling, possibly staring at a person [box0] in the restaurant
**Ferret (Ours):**
The man [box0] is having a meal at a table [box1]. In front of him, there is a plate [box2] with a variety of food items such as fruit and eggs. There is also a glass [box3] filled with water and a bottle [box4].

