# OpenReview forum: "Ferret: Refer and Ground Anything Anywhere at Any Granularity"
_ICLR.cc/2024/Conference — ICLR 2024 spotlight_

### Official Review · Reviewer_dfTo · 2023-10-27

**Soundness:** 3 good
**Presentation:** 3 good
**Contribution:** 3 good
**Rating:** 6
**Confidence:** 4

**Summary:**

This paper proposes a large multimodal model Ferret, which, compared with prior works, is especially good at referring and grounding. In Ferret, they propose a hybrid region representation and a novel spatial-aware visual sampler to represent the visual and regional input. To train Ferret, they creat GRIT, a large-scale ground-and-refer instruction tuning dataset. They also introduce Ferret-Bench to evaluate the grounding, referring, and reasoning capabilities of LMMs.

**Strengths:**

1. The proposed GRIT dataset is meaningful to the vision and language research.
2. The proposed Spatial-Aware Visual Sampler and Hybrid Region Representation are well-motivated.
3. The experiment results show the better capabilities of the trained model on multiple referring and grounding tasks and validate the effectiveness of the spatial-aware visual sampler module.

**Weaknesses:**

1. The ablation on hybrid region representation is missing.
2. Not a strong weakness, but whether the model performs well on non-referring or grounding tasks needs more validation. E.g. VQA_v2, MME, general captioning, etc. And it seems the caption evaluation is not as good as InstructBLIP.

**Questions:**

1. Is the evaluation on Flickr30k grounded caption fine-tuned or zero-shot?

---

> ### Author Response · Authors · 2023-11-18
> **Author Rebuttal**
>
> > **W1:The ablation on hybrid region representation is missing.**
>
> **Answer:** Thank you for your suggestion. Here, we added the ablation experiment of hybrid region representation on LVIS Referring Object Classification.
>
> | Region                       | Point | Box  | Free-Form |
> |------------------------------|-------|------|-----------|
> | Hybrid Region Representation | 65.5  | 76.2 | 69.0      |
> | Coordinate Only              | 64.6  | 74.5 | -         |
> | Region Only                  | 60.4  | 70.1 | 64.2      |
>
> As we can see, compared with the region-only method, the coordinate-only method can bring better performance in point and box referring but theoretically cannot handle free-form shape. When combining those two into a hybrid region representation, the performance on three types of referring gets further boosted.
>
> > **W2: Not a strong weakness, but whether the model performs well on non-referring or grounding tasks needs more validation. E.g. VQAv2, MME, general captioning, etc. And it seems the caption evaluation is not as good as InstructBLIP.**
>
> **Answer:** We appreciate the reviewer's insight and recognize its significance in the broader context of model evaluation.  Our research is primarily focused on enabling referring and grounding capabilities in MLLM. This focus has informed the design and optimization of our model, leading us to build datasets and design evaluation metrics that directly align with these specific areas. We indeed also reported Ferret's superior results on POPE and LLaVA-Bench that are non-referring or grounding tasks, but we also understand your suggestion about broadening our evaluation scope.
>
> As for more general-purpose evaluations, such as VQAv2 and general captioning, most previous works directly add VQA data and COCO captioning data in their instruction-tuning stage (Shikra and InstructBLIP), which makes Ferret unable to fairly compare with them. As for more all-around evaluations (MME, MM-Vet, etc),  they actually examine multiple specific domain knowledge, such as OCR and commonsense reasoning. Previous works (LLaVA-1.5, InstructBLIP) directly add OCR-VQA and OK-VQA data in their training, which also makes Ferret unable to fairly compare.
>
> Nevertheless, we agree with the reviewer about the trend of building an all-around model. As an ongoing future work, we want to rigidly study the effects of referring/grounding in general evaluation benchmarks, which needs significantly more effort to get some of the results ready. Also, we are indeed actively expanding our model’s capabilities to more universal and general scenarios, and studying how referring/grounding can better help or be combined with those tasks not simply from performance numbers. We appreciate the reviewer’s suggestion and feedback again.
>
> > **Q1:Is the evaluation on Flickr30k grounded caption fine-tuned or zero-shot?**
>
> **Answer:**  It's not considered as zero-shot because around 30\% of the Flickr30k training data has been seen in our GRIT dataset, essentially guiding the model to emulate the style of the output. The other SoTA methods we mentioned in the table about Flickr30k grounded caption (Tab.2 in the manuscript) utilize the whole training set of Flickr30k, while our Ferret still outperforms them.

---

> ### Author Response · Authors · 2023-11-22
> **Reviewer-Author Discussion Period Ends in ONE Day**
>
> Dear Reviewer dfTo,
>
> Thank you again for your insightful reviews of our submission. Following your feedback, we have provided a detailed response trying to address the concerns you raised. As the deadline is approaching, it would be very helpful if you could revisit our clarifications and let us know if any ambiguities remain before the reviewer-author discussion period ends. We would greatly appreciate any further comments you might have regarding our work, and we are fully committed to answering any questions.
>
> Your effort and time in reviewing our submission are sincerely appreciated.
>
> Warm regards,
> Author(s)

---

> ### Author Response · Authors · 2023-11-23
> **Reviewer-Author Discussion Period Ends in 7 HOURS**
>
> Dear Reviewer dfTo,
>
> The reviewer-author discussion period is going to end in less than 7 hours. We are eager to hear your feedback and comments on our responses. Don't hesitate to let us know your thoughts. Thank you again for your efforts and time in reviewing our submission.
>
> Best Regards,\
> Author(s)

---

### Official Review · Reviewer_6Qgf · 2023-10-31

**Soundness:** 2 fair
**Presentation:** 3 good
**Contribution:** 3 good
**Rating:** 6
**Confidence:** 4

**Summary:**

This paper introduces Ferret, a Multimodal Large Language Model (MLLM) capable of understanding spatial referring within an image and grounding open-vocabulary descriptions. The paper also introduces the dataset GRIT with 1.1M samples and an additional 130K hard negative data.

Ferret includes several components:
1. A powerful hybrid region representation integrating discrete coordinates and continuous features to represent image regions.
2. A spatial-aware visual sampler for handling various region shapes and extracting continuous features.
3. Integration with LLM for referring and grounding tasks.

Ferret achieves superior performance in classical referring and grounding tasks and outperforms existing methods and it shows improved capability in describing image details and reduces object hallucination.

**Strengths:**

1. The paper is presented very well.
2. The paper shows a reasonable motivation that humans inherently possess the ability to learn from one task and generalize to another between referring and grounding. This underscores the essential need to unify referring and grounding processes.
3. The hybrid region representation and spatial-aware visual sampler make the framework flexible to take different form of region definition.
4. The framework shows a good way of utilization of Large Language Model.
5. Contribution of the dataset.

**Weaknesses:**

1. No open source code for the code and dataset. I would raise the soundness score if code and dataset are open, either attached in the supplementary or released in the public repo.
2. The hierarchy of the dataset is a bit complicated. This may not be practical for costume dataset.
3. Very engineering paper, extensive work, but not much scientific novelty.

**Questions:**

1. Table 5 shows that mutual benefits of grounding and referring. From the results, it seems grounding task can help more for referring task than the other way around. How to interpret this effect?
2. For section 4.2, how to evaluate quality of the generated data from ChatGPT and GPT4?
3. You mensioned in the Ferret-Bench is via GPT4 as a judge. But some of the data are collected from GPT4, is a reason it is better than all other models in Table 7.
4. Can chatgpt take multimodal input? Since when you collecting dataset via LLM, the author mentioned they use ChatGPT first and then use GPT-4 to refine it. I am wondering how ChatGPT can take image as input.

It is extensive of work. I would like to raise my score if the questions are addressed and the code and data are public.

---

> ### Author Response · Authors · 2023-11-18
> **Author Rebuttal (1/3)**
>
> > **W1: No open source code for the code and dataset. I would raise the soundness score if code and dataset are open, either attached in the supplementary or released in the public repo.**
>
> **Answer:** For the sake of the double-blind policy, we create a **public anonymous repo** in [https://anonymous.4open.science/r/ferret-anonymous-6773/](https://anonymous.4open.science/r/ferret-anonymous-6773/), and post all our training/evaluation/serving code, Ferret-Bench data, and a small subset of training data for preview. We will open-source the full training data and pre-trained model checkpoints upon our institution's approval.
>
> > **W2:The hierarchy of the dataset is a bit complicated. This may not be practical for costume dataset.**
>
> **Answer:** Thank you for your insightful comment. We believe the hierarchical nature of the dataset is reflective of the inherent complexity in the real world. In the realm of computer vision, this hierarchy is fundamental. It begins with basic elements such as object detection [1] and segmentation [2], progresses to understanding relationships between these objects[3], and culminates in the comprehension of complex scenes through scene graphs[4].
>
> Our model, Ferret, is designed to leverage this hierarchical structure, thereby enhancing its robustness and adaptability across various contexts. For finetuning using custom datasets, it's not necessary for these datasets to have a hierarchical structure. You have the flexibility to fine-tune the model with your preferred data. We recommend two viable strategies: (1). fine-tuning our pre-trained Ferret model with your custom dataset; or (2) incorporating your dataset into the initial training phase alongside our existing data.
>
> [1] Girshick, Ross. "Fast r-cnn." Proceedings of the IEEE international conference on computer vision. 2015. \
> [2] He, Kaiming, et al. "Mask r-cnn." Proceedings of the IEEE international conference on computer vision. 2017. \
> [3] Lu C, Krishna R, Bernstein M, Fei-Fei L. Visual relationship detection with language priors. InComputer Vision–ECCV 2016: 14th European Conference, Amsterdam, The Netherlands, October 11–14, 2016, Proceedings, Part I 14 2016 (pp. 852-869). Springer International Publishing. \
> [4] Xu D, Zhu Y, Choy CB, Fei-Fei L. Scene graph generation by iterative message passing. InProceedings of the IEEE conference on computer vision and pattern recognition 2017 (pp. 5410-5419).
>
> > **W3:Very engineering paper, extensive work, but not much scientific novelty.**
>
> **Answer:** Thank you for recognizing the extensive amount of work and contribution of our paper. We appreciate your feedback and would like to address your concerns regarding the novelty of our work.
>
> We want to clarify that the Spatial-aware Visual Sampler and the hybrid region representation should be novel solutions to deal with arbitrary referring regions in MLLM.
>
> Also, we believe that scientific novelty is going beyond just modeling, as it encompasses a broad spectrum covering data, modeling, training recipes, and evaluation, etc.  In our work, besides Spatial-aware Visual Sampler and the hybrid region representation, how we prepare the hierarchical and robust refer-and-ground instruction-tuning data, and our proposed Ferret-Bench as the new and advanced benchmark for this domain should arguably also be considered as novelties.

---

> ### Author Response · Authors · 2023-11-18
> **Author Rebuttal (2/3)**
>
> > **Q1:Table 5 shows that mutual benefits of grounding and referring. From the results, it seems grounding task can help more for referring task than the other way around. How to interpret this effect?**
>
> **Answer:** Thank you for the brilliant observation. We consider the following two factors:
>
> * The absolute values of performance gains on different datasets cannot be treated in the same way.
>     * On the one hand, referring and grounding tasks are measured in different metrics. Same as common practice, referring is evaluated by accuracy, while Flickr30k grounding is evaluated by Recall@1.
>     * On the other hand, in some datasets, such as Flickr30k, the baselines can already achieve high scores, and thus, further enhancement appears relatively modest in terms of values. For a straightforward example, improving 10\% on top of 80\% is much more difficult than over 50\%.
>
> * Difference in the amount of training data. Our GRIT dataset contains a larger quantity of grounding data compared to referring data. This imbalance in data volume may contribute to the observed different benefits for each task type.
>
> > **Q2: For section 4.2, how to evaluate quality of the generated data from ChatGPT and GPT4?**
>
> **Answer:** This is a great question. We consider the evidence from the following three aspects.
>
> * We post several randomly selected examples from our GPT-generated data in the anonymous repo ([https://anonymous.4open.science/r/ferret-anonymous-6773/training_data.md](https://anonymous.4open.science/r/ferret-anonymous-6773/training_data.md)). It shows satisfactory quality in terms of accurate referring and grounding, and correct semantics and reasoning.
>
> * Generating visual instruction tuning data through GPT has been used in many previous works, such as LLaVA, and is proven to be of satisfactory quality.  In our collection process, we even provide more accurate visual symbolic conditions including relationships and region descriptions, besides only objects and global captions as in LLaVA. Therefore, the quality of our data should also be guaranteed.
>
> * We further provide a human evaluation of the quality of our GPT-generated data through Amazon Mechanical Turk. To be specific, inspired by the GPT generation quality evaluation in NLP [1] [2], we introduced the following four criteria:
>     * **Localization Precision**: Whether the regions mentioned in the text precisely ground to the correct regions in the image.
>     * **Honesty to image**: Whether the conversations are honest to the facts in the image.
>     * **Helpfulness**:  Whether the answers are correct for the questions.
>     * **Harmlessness**: Whether the conversations do not have harm to humans. If it has hate speech or promotes violence, it's not harmless.
>
>     Every criterion can be scored from 1 to 5, where a higher score means better quality.
>
>     We randomly select 100 samples from our GPT-assisted visual instruction data for human evaluation. Each sample is required to be rated 3 times by different workers from Amazon Mechanical Turk. In total, we obtained 300 ratings, and we further removed some outlier assessments where three workers had a large disagreement for the same sample, which gave us 267 ratings in the end.  We average the scores of all those ratings and show the statistics of our human evaluation results in the following table. As we can see, in general, the quality of our data is satisfactory, especially in localization precision, helpfulness, and harmlessness. The score of honesty to image is relatively lower by a small margin, which might be due to the GPT's slight hallucination.
>
>     |                     | Localization Precision | Honesty to image | Helpfulness | Harmlessness |
>     |---------------------|------------------------|------------------|-------------|--------------|
>     | Average Score (1-5) | 4.42                   | 4.29             | 4.41        | 4.72         |
>
>     [1]. Peng, Baolin, et al. "Instruction tuning with gpt-4." arXiv preprint arXiv:2304.03277 (2023).
>
>     [2]. Askell, Amanda, et al. "A general language assistant as a laboratory for alignment." arXiv preprint arXiv:2112.00861 (2021).

---

> ### Author Response · Authors · 2023-11-18
> **Author Rebuttal (3/3)**
>
> > **Q3: You mensioned in the Ferret-Bench is via GPT4 as a judge. But some of the data are collected from GPT4, is a reason it is better than all other models in Table 7.**
>
> **Answer:** Thank you for mentioning this concern. To clarify, all other models in Tab. 7 also use data collected from ChatGPT/GPT-4, such as LLaVA's instruction data and Shikra's instruction data. Since the tasks in Ferret-Bench (Tab. 7) require a joint capability of accurate referring, grounding, and correct reasoning, we hypothesize the following two reasons that stand Ferret out compared with other models:
>
> * The basic referring and grounding capabilities of Ferret are stronger than other models, which can be observed in Tab. 1, 2, and 3 of our submission. It establishes a better base for the tasks in Ferret-Bench.
>
> * Better GPT-generated data. On the one hand, our data is conditioned on more fine-grained visual symbolic information (relationships and region descriptions), which provides more details. On the other hand, our GPT-generated data is prompted to be more aware of the joint referring/grounding and reasoning, which is of a higher quality for complex scenarios.
>
> > **Q4: Can chatgpt take multimodal input? Since when you collecting dataset via LLM, the author mentioned they use ChatGPT first and then use GPT-4 to refine it. I am wondering how ChatGPT can take image as input.**
>
> **Answer:**  In LLaVA's instruction data collection, for each image, its ground-truth bounding boxes (w/ coordinates) and five ground-truth captions are provided to ChatGPT as the textual representation of the image. Although it might not cover all the details of the images, it's sufficient for GPT to generate precise instruction-tuning data by utilizing the commonsense knowledge learned in the language domain. This method has proven to be effective in LLaVA and is widely used in recent MLLMs.
>
> In our GRIT data collection, besides following LLaVA's setup, as we stated in Sec. 3.2 of our submission, `Besides objects and global captions usually used as before, our visual symbolic context additionally includes physical relationships between objects and region captions along with coordinates of them.' This provides more fine-grained information in the image and helps the GPT to generate higher-quality data. Two examples can be found in Tab. 11 and Tab. 13 of the manuscript. The detailed prompts are shown in Tab. 10 and Tab 12.

---

> ### Author Response · Authors · 2023-11-22
> **Reviewer-Author Discussion Period Ends in ONE Day**
>
> Dear Reviewer 6Qgf,
>
> Thank you again for your insightful reviews of our submission. Following your feedback, we have provided a detailed response trying to address the concerns you raised. As the deadline is approaching, it would be very helpful if you could revisit our clarifications and let us know if any ambiguities remain before the reviewer-author discussion period ends. We would greatly appreciate any further comments you might have regarding our work, and we are fully committed to answering any questions.
>
> Your effort and time in reviewing our submission are sincerely appreciated.
>
> Warm regards, \
> Author(s)

---

> ### Author Response · Authors · 2023-11-23
> **Reviewer-Author Discussion Period Ends in 7 HOURS**
>
> Dear Reviewer 6Qgf,
>
> The reviewer-author discussion period is going to end in less than 7 hours. We are eager to hear your feedback and comments on our responses. Don't hesitate to let us know your thoughts. Thank you again for your efforts and time in reviewing our submission.
>
> Best Regards,\
> Author(s)

---

### Official Review · Reviewer_7dkr · 2023-11-01

**Soundness:** 3 good
**Presentation:** 3 good
**Contribution:** 3 good
**Rating:** 8
**Confidence:** 4

**Summary:**

The paper tackles the challenge of training a Multi-modal Large Language Model to accurately interpret input visual references, such as points, bounding boxes, free-form shapes, with respect to the image (referring) and ground the output text to relevant image regions (grounding). The authors propose a unified framework, called Ferret, for jointly solving the visual referring and grounding problem. They provide a new curated dataset (GRIT) that consists of existing and newly collected data for training, as well as a new benchmark (Ferret-Bench). In the evaluation section, the authors show that the proposed method either exceeds (on the Ferret-Bench and grounded captioning) or is on par with the concurrent SOTA methods such as Shikra. In addition, the proposed method can accept a variety of user input on images as part of referring expressions, including scribble and freeform shapes, in addition to the traditional points and bounding boxes. However, the way how these different visual reference types are processed is conceptually similar to the Visual Sampler proposed in SEEM (Zou et al., 2023), and performs similarly despite the added complexity.

**Strengths:**

S1. This seems to be one of the first MLLMs to support a variety of visual reference types, such as point, box, scribble, polygons, and masks.

S2. The authors provide a curated dataset called GRIT that consists of existing datasets and newly collected data for training MLLMs with visual referring and grounding capabilities.

S3.  The authors provide a new benchmark, Ferret-Bench, which covers two new types of evaluation task for visual referencing (description and reasoning) in addition to the conversation grounding task. The key difference to existing benchmarks such as RefCOCO+, RefCOCOg, or PointQA [a] is that the questions include visual references (in forms of bounding boxes). For example, “What is the purpose of the object [x1 y1 x2 y2]?”
 - [a] Point and Ask: Incorporating Pointing into Visual Question Answering, Mani et al., 2022

S4. The paper provides comparison with the SOTA methods and the concurrent methods in the evaluation section.

**Weaknesses:**

W1. The paper omits any discussion on the limitations or potential failure scenarios of the proposed method.

W2. The significance of the proposed Spatial-Aware Visual Sampler is minimal. The idea of sampling the visual features over the grid is in the same spirit as the Visual Sampler in SEEM (Zou et al., 2023), although the details of how the points features are aggregated and pooled are different. Performance-wise, the Spatial-Aware Visual Sampler is shown to be only marginally better than the Visual Sampler in SEEM as shown in the ablation study section.

W3. While the idea of jointly solving referring (with explicit visual cues, such as markings on the image) and grounding in one unified framework makes sense as the two tasks are interrelated, this idea was also explored in concurrent works (Chen et al., 2023a) and (Peng et al., 2023).

**Questions:**

Q1. It appears that the proposed method significantly outperforms Shikra on Ferret-Bench, but not much on existing datasets. I wonder if the authors can explain why.

Q2. I think the quality of writing could be further improved. For example, it is not clear to me what the authors are trying to imply by “First of all, we choose MLLM as the bedrock of Ferret due to their powerful vision-language global understanding capability”. I am guessing that they wanted to say that the Ferret is built on top of existing MLLMs to leverage their powerful vision-language capability?

---

> ### Author Response · Authors · 2023-11-18
> **Author Rebuttal (1/2)**
>
> > **W1: The paper omits any discussion on the limitations or potential failure scenarios of the proposed method.**
>
> **Answer:** Thank you for your valuable feedback. Here is our discussion about limitations and potential failure scenarios.  We also updated it in the appendix of the manuscript, highlighted in blue for easy reference.
>
> Failure Scenarios: (1). Referring to too many objects (more than 3) in one question might not be as accurate as referring to each of them in separate conversations. This is likely due to a relative scarcity of training data that mentions too many objects. (2). The referring and grounding of very small objects is less accurate than large or medium objects. It's a common challenge in object detection. However, we think further improving input image resolution is able to help.
>
> Limitations: (1). Not good at other languages because the training dataset is curated only in English. Although Ferret shows some emergent referring and grounding capability in other languages, its performance in other languages is still worse than in English. Future incorporation of multilingual training data could potentially mitigate this. (2). Similar to many large language models, Ferret has the potential to generate harmful or factually incorrect responses.  (3). Ferret is not designed for segmentation tasks requiring mask outputs.
>
> > **W2. The significance of the proposed Spatial-Aware Visual Sampler is minimal. The idea of sampling the visual features over the grid is in the same spirit as the Visual Sampler in SEEM (Zou et al., 2023), although the details of how the points features are aggregated and pooled are different. Performance-wise, the Spatial-Aware Visual Sampler is shown to be only marginally better than the Visual Sampler in SEEM as shown in the ablation study section.**
>
> **Answer:** We appreciate the reviewer's concerns regarding the significance of our Spatial-Aware Visual Sampler. To address this, we refer to the experimental results presented in Tab. 6 of our submission, which highlight the ablation across different types of regions. For small and simple regions such as circles derived from points, we observe a modest improvement of 0.6\% in point referring tasks. While for larger and more complex shapes like boxes or free-form regions, the Spatial-Aware Visual Sampler shows more significant improvement, ranging from 1\% to 2\%.
>
> The above observation makes sense, as when the region is small, simple average pooling in SEEM might already be enough. Although the enhancements observed in these smaller regions seem modest, our approach demonstrates substantial benefits when addressing the intricacies of larger regions.

---

> > ### Comment · Reviewer_7dkr · 2023-11-22
> > **RE: Rebuttal (1/2)**
> >
> > - W1: Thanks for providing limitations and failure cases. I recommend also including examples in the appendix.
> > - W2: I appreciate breaking down the results, but I am not convinced about labeling 1-2% improvement as "significant". I still think it is a good paper, so I think there's no need for the authors to overemphasize every aspect.
> >
> > [Important] I just noticed that the metrics are not specified in the captions for the tables (Table 1, 5,6), so I strongly recommend adding them in the final manuscript.

---

> ### Author Response · Authors · 2023-11-18
> **Author Rebuttal (2/2)**
>
> > **W3. While the idea of jointly solving referring (with explicit visual cues, such as markings on the image) and grounding in one unified framework makes sense as the two tasks are interrelated, this idea was also explored in concurrent works (Chen et al., 2023a) and (Peng et al., 2023).**
>
> **Answer:** Thank you for your comments. We appreciate the opportunity to clarify the unique aspects and contributions of our work in this domain.
>
> In the original manuscript, we mentioned and compared these two concurrent works in related works. To clarify our unique positioning, we also post a detailed comparison table below and in the appendix of newly-uploaded manuscript (highlighted in blue). Additionally, we would like to highlight one more distinction: we demonstrated the mutual benefits of referring and grounding through detailed ablation studies in Table 5 of our submission.
>
>
> | Model       | Input Types |    |    | Output Grounding | Data Construction |    |    | Quantitative Eval. of Refer/Ground w. Chat|
> |-------------|-------------|----|----|------------------|-------------------|----|----|--------------------|
> |             | Point       | Box | Free-form |                  | Convention       | GPT-Generate | Robustness |                    |
> | BuboGPT     | ❌          | ❌   | ❌         | ✅                | ✅                 | ❌           | ❌         | ❌                  |
> | Vision-LLM  | ❌          | ❌   | ❌         | ✅                | ✅                 | ❌           | ❌         | ❌                  |
> | Kosmos-2    | ❌          | ✅   | ❌         | ✅                | ✅                 | ❌           | ❌         | ❌                  |
> | Shikra      | ✅          | ✅   | ❌         | ✅                | ✅                 | ✅           | ❌         | ❌                  |
> | GPT4-ROI    | ❌          | ✅   | ❌         | ❌                | ✅                 | ❌           | ❌         | ❌                  |
> | PVIT        | ❌          | ✅   | ❌         | ❌                | ✅                 | ✅           | ❌         | ✅                  |
> | **Ferret**  | ✅          | ✅   | ✅         | ✅                | ✅                 | ✅           | ✅         | ✅                  |
>
> *Note: `Convention` refers to a comprehensive collection of publicly available data that has been transformed using templates, `GPT-Generate` signifies the generated refer/ground datasets employing GPT, and `Robustness` denotes datasets aimed at mitigating hallucination and improving robustness.*
>
> > **Q1. It appears that the proposed method significantly outperforms Shikra on Ferret-Bench, but not much on existing datasets. I wonder if the authors can explain why.**
>
> **Answer:** Thank you for your insightful observation. First of all, in Ferret-Bench, Ferret can outperform Shikra much more in referring-related tasks than grounding tasks. This is coherent with the performance comparison of Ferret and Shikra in existing data.
>
> Regarding why Ferret can outperform Shikra more in general than existing works, besides Ferret can already outperform Shikra in basic referring/grounding capabilities, the larger difference may be credited to the fact that Ferret integrates referring and grounding with semantics, knowledge, and reasoning in a global context better. GPT-4 data, hierarchical data collection, and our model design may be the effective factors. The other possible reason is that the performance of the existing benchmarks may already be close to saturation. Therefore, we recommend adopting Ferret-Bench, which covers more complex referring and grounding scenarios, as a new benchmark for future research in this domain.
>
> > **Q2. I think the quality of writing could be further improved. For example, it is not clear to me what the authors are trying to imply by “First of all, we choose MLLM as the bedrock of Ferret due to their powerful vision-language global understanding capability”. I am guessing that they wanted to say that the Ferret is built on top of existing MLLMs to leverage their powerful vision-language capability?**
>
> **Answer:** Thank you so much for your suggestion. We have modified the corresponding sentence to `First of all, we choose MLLM as the bedrock of Ferret to leverage their powerful vision-language understanding capability`. Additionally, we also revised several other sentences to improve the quality of writing (highlighted in blue in the new manuscript).

---

> > ### Comment · Reviewer_7dkr · 2023-11-22
> > **RE: Rebuttal (2/2)**
> >
> > - W3. This table is great. Thanks for adding it to the manuscript.
> > - W4.
> > > the larger difference may be credited to the fact that Ferret integrates referring and grounding with semantics, knowledge, and reasoning in a global context better. GPT-4 data, hierarchical data collection, and our model design may be the effective factors.
> >
> > I think the authors' answer for this is vague and unscientific. I suggest pinpointing the primary factor responsible for the performance gap and incorporating this clarification into the final manuscript.

---

> > > ### Author Response · Authors · 2023-11-23
> > > **Thank you for your response.**
> > >
> > > Dear Reviewer 7dkr,
> > >
> > > We really appreciate your valuable feedback, as well as your effort and time in reviewing our paper. Here is our follow-up response to your newest comments.
> > >
> > > > **1.  Thanks for providing limitations and failure cases. I recommend also including examples in the appendix.**
> > >
> > > **Answer**: Thank you for the valuable suggestion. We will include more visualization results in the appendix to provide clearer insights into the limitations and failure cases.
> > >
> > > > **2: I appreciate breaking down the results, but I am not convinced about labeling 1-2% improvement as "significant". I still think it is a good paper, so I think there's no need for the authors to overemphasize every aspect.\
> > > [Important] I just noticed that the metrics are not specified in the captions for the tables (Table 1, 5,6), so I strongly recommend adding them in the final manuscript.**
> > >
> > > **Answer**: We acknowledge your concern regarding the characterization of a 1-2% improvement as 'significant.' We will revise the language to more accurately reflect the impact of these improvements. Additionally, we have added the evaluation metrics in the captions of Tables 1, 5, and 6 in the newly-uploaded manuscript (highlighted in blue), as recommended.
> > >
> > > > **3:  This table is great. Thanks for adding it to the manuscript.**
> > >
> > > **Answer**: We are pleased to hear that the addition of the table was well-received. Thank you for your positive feedback!
> > >
> > > > **4. `the larger difference may be credited to the fact that Ferret integrates referring and grounding with semantics, knowledge, and reasoning in a global context better. GPT-4 data, hierarchical data collection, and our model design may be the effective factors.` \
> > > I think the authors' answer for this is vague and unscientific. I suggest pinpointing the primary factor responsible for the performance gap and incorporating this clarification into the final manuscript.**
> > >
> > > **Answer**: Thank you for pointing out the need for more specificity in our explanation. As we stated, many conventional benchmarks are close to saturated, and there is still a lot of space for improvement in Ferret-Bench. Regarding our model, one potential hypothesis is that our hierarchical and robust dataset may play the primary factor in boosting the performance. Starting from this hypothesis, we will conduct further analysis on each design choice of the dataset, and also consider model design, specifically Spatial-aware Visual Sampler, as a potential minor factor to be validated. We will update the analysis in the final manuscript.

---

> ### Author Response · Authors · 2023-11-22
> **Reviewer-Author Discussion Period Ends in ONE Day**
>
> Dear Reviewer 7dkr,
>
> Thank you again for your insightful reviews of our submission. Following your feedback, we have provided a detailed response trying to address the concerns you raised. As the deadline is approaching, it would be very helpful if you could revisit our clarifications and let us know if any ambiguities remain before the reviewer-author discussion period ends. We would greatly appreciate any further comments you might have regarding our work, and we are fully committed to answering any questions.
>
> Your effort and time in reviewing our submission are sincerely appreciated.
>
> Warm regards,\
> Author(s)

---

### Meta-Review · Area_Chair_pfTj · 2023-12-06

**Metareview:**

Paper summary
- The paper proposes Ferret, a multi-modal large language model (MLLM) that allows for visual grounding of text into images at different granularities (point, box, arbitrarily shaped region).  The regions are encoded using discrete-continuous hybrid representation for encoding regions as combination of discrete coordinates and extracted visual features based on a mask.  To handle the arbitrarily shaped regions, a sampling approach is used to sample features from a dense feature grid using the mask.  To allow for a variety of grounding task, the model takes as input a visual image, and text that is augmented with references to visual regions (by adding the region coordinates and place holder for continuous visual features for the region as tokens).  The model is built on top of a LLM (i.e. the tokenized input is passed into the LLM and the LLM is fine-tuned using a instruction tuning dataset to produce appropriate output) and can support tasks such as visual grounding, captioning of regions, and multi-modal chatting.  To fine-tune the LLM, an instruction-tuning dataset (GRIT) of 1.1 samples was collected.  Experiments show that proposed model can outperform prior work on referring object classification, visual grounding, grounded captioning, and multimodal chatting.


The contributions of this work includes
- GRIT: a instruction tuning dataset for visual reference and grounding.
- New benchmark (Ferret-Bench) for evaluating different models' ability to perform multi-modal chatting
- Unified model to handle a variety of grounding tasks with hybrid representation to encode different visual regions
- Experiments comparing the proposed method to prior work, and ablation studies.

Strengths
- Flexible approach to visual grounding tasks that can handle a variety of visual references (point, box, scribble, polygons, masks)
- Contributions of dataset (GRIT) and Ferret-Bench will be useful for the community
- Experiments show the proposed approach outperforms prior work

**Justification For Why Not Higher Score:**

- Reviewers noted the similarity of the proposed model to SEEM [Zou et al. 2023]

**Justification For Why Not Lower Score:**

- The paper received positive rating from all reviewers.  The AC believe the flexibility offered by the representation and the unified task architecture would be of interest to the vision-language research community and should be highlighted.

---

### Decision · Program_Chairs · 2024-01-16

Accept (spotlight)